# Sphingosine 1-phosphate-regulated transcriptomes in heterogenous arterial and lymphatic endothelium of the aorta

Eric Engelbrecht[1†], Michel V Levesque[1†], Liqun He[2,3], Michael Vanlandewijck[2,3], Anja Nitzsche[4], Hira Niazi[4], Andrew Kuo[1], Sasha A Singh[5], Masanori Aikawa[5], Kristina Holton[6], Richard L Proia[7], Mari Kono[7], William T Pu[8,9], Eric Camerer[4], Christer Betsholtz[2,3], Timothy Hla[1]*

[1]Vascular Biology Program, Boston Children's Hospital, Deapartment of Surgery, Harvard Medical School, Boston, United States; [2]Department of Immunology, Genetics and Pathology, Rudbeck Laboratory, Uppsala University, Uppsala, Sweden; [3]Karolinska Institutet/AstraZeneca Integrated Cardio Metabolic Centre (KI/AZ ICMC), Karolinska Institutet, Huddinge, Sweden; [4]Université de Paris, INSERM U970, Paris Cardiovascular Research Center, Paris, France; [5]Center for Interdisciplinary Cardiovascular Sciences, Department of Medicine, Brigham and Women's Hospital, Harvard Medical School, Boston, United States; [6]Harvard Medical School Research Computing, Boston, United States; [7]Genetics of Development and Disease Branch, National Institute of Diabetes and Digestive and Kidney Diseases, National Institutes of Health, Bethesda, United States; [8]Department of Cardiology, Boston Children's Hospital, Harvard Medical School, Boston, United States; [9]Harvard Stem Cell Institute, Harvard University, Cambridge, United States

*For correspondence:
timothy.hla@childrens.harvard.edu

[†]These authors contributed equally to this work

**Abstract** Despite the medical importance of G protein-coupled receptors (GPCRs), in vivo cellular heterogeneity of GPCR signaling and downstream transcriptional responses are not understood. We report the comprehensive characterization of transcriptomes (bulk and single-cell) and chromatin domains regulated by sphingosine 1-phosphate receptor-1 (S1PR1) in adult mouse aortic endothelial cells. First, S1PR1 regulates NFκB and nuclear glucocorticoid receptor pathways to suppress inflammation-related mRNAs. Second, S1PR1 signaling in the heterogenous endothelial cell (EC) subtypes occurs at spatially-distinct areas of the aorta. For example, a transcriptomically distinct arterial EC population at vascular branch points (aEC1) exhibits ligand-independent S1PR1/ß-arrestin coupling. In contrast, circulatory S1P-dependent S1PR1/ß-arrestin coupling was observed in non-branch point aEC2 cells that exhibit an inflammatory gene expression signature. Moreover, S1P/S1PR1 signaling regulates the expression of lymphangiogenic and inflammation-related transcripts in an adventitial lymphatic EC (LEC) population in a ligand-dependent manner. These insights add resolution to existing concepts of endothelial heterogeneity, GPCR signaling and S1P biology.

## Introduction

Sphingosine 1-phosphate (S1P), a circulating lipid mediator, acts on G protein-coupled S1P receptors (S1PRs) to regulate a variety of organ systems. S1PR1, abundantly expressed by vascular endothelial cells (ECs), responds to both circulating and locally-produced S1P to regulate vascular development, endothelial barrier function, vasodilatation and inflammation (*Proia and Hla, 2015*).

S1P binding to S1PR1 activates heterotrimeric $G_{\alpha i/o}$ proteins, which regulate downstream signaling molecules such as protein kinases, the small GTPase RAC1, and other effector molecules to influence cell behaviors such as shape, migration, adhesion and cell-cell interactions. Even though S1PR1 signaling is thought to evoke transcriptional responses that couple rapid signal transduction events to long-term changes in cell behavior, such mechanisms are poorly understood, especially in the vascular system.

Subsequent to activation of $G_{\alpha i/o}$ proteins and RAC1, the S1PR1 C-terminal tail gets phosphorylated and binds to ß-arrestin, leading to receptor desensitization and endocytosis (*Liu et al., 1999*; *Oo et al., 2007*). While S1PR1 can be recycled back to the cell surface for subsequent signaling, sustained receptor internalization brought about by supra-physiological S1P stimulation or functional antagonists that are in therapeutic use leads to recruitment of WWP2 ubiquitin ligase and lysosomal/proteasomal degradation of the receptor (*Oo et al., 2011*). Thus, ß-arrestin coupling down-regulates S1PR1 signals. However, studies of other GPCRs suggest that ß-arrestin coupling can lead to biased signaling distinct from $G_{\alpha i/o}$ -regulated events (*Wisler et al., 2018*). Distinct transcriptional changes brought about by $G_{\alpha i/o}$ – and ß-arrestin-dependent pathways are not known.

Studies of ß-arrestin and RAC1 knockout mice highlighted the unique physiological functions of these proteins in S1PR1 signaling. Deletion of *S1pr1* or *Rac1* in endothelium results in lethality at embryonic day (E)13.5 and E9.5, respectively (*Allende et al., 2003*; *Tan et al., 2008*). In contrast, mice with germline null alleles for *barr1* (*Conner et al., 1997*) or *barr2* (*Bohn et al., 1999*) survive without gross abnormalities while *barr1*$^{-/-}$*barr2*$^{-/-}$ mice survive to term (*Zhang et al., 2010*).

Our understanding of GPCR signaling in vivo, particularly with respect to direct transcriptional targets and spatial specificity of signaling, is limited. To address this, *Kono et al. (2014)* developed S1PR1 reporter mice (S1PR1-GS mice) which record receptor activation at single-cell resolution (*Kono et al., 2014*; *Barnea et al., 2008*). S1PR1-GS mice harbor one wild-type *S1pr1* allele and one targeted knock-in allele, which encodes S1PR1-tTA and ß-arrestin-TEV protease fusion proteins separated by an internal ribosome entry sequence (*Kono et al., 2014*). Breeding the *S1pr1* knock-in allele with the tTA-responsive *H2B-GFP* allele generates an S1PR1-GS mouse. In S1PR1-GS mice, the β-arrestin-TEV fusion protein triggers release of tTA from the C terminus of modified S1PR1 when β-arrestin-TEV and S1PR1-tTA are in close proximity. Free tTA enters the nucleus and activates *H2B-GFP* reporter gene expression. Since S1PR1-GS mouse embryonic fibroblast cells respond to S1P with an $EC_{50}$ of 43 nM (*Kono et al., 2014*; *Lee et al., 1998*), the S1PR1 reporter system accurately reports S1PR1 activation. Indeed, structural and functional analyses of other GPCRs suggest that the C-terminal phosphorylation patterns determine the strength of ß-arrestin binding, which was accurately predicted by the ß-arrestin coupling (*Zhou et al., 2017*). The in vivo half-life of H2B-GFP protein is ~24 days in hair follicle stem cells (*Waghmare et al., 2008*). Therefore, GFP expression in this reporter mouse represents the cumulative record of S1PR1 activation in vivo.

We previously showed that high levels of endothelial GFP expression (i.e. S1PR1/ß-arrestin coupling) are prominent at the lesser curvature of the aortic arch and the orifices of intercostal branch points (*Galvani et al., 2015*). In addition, inflammatory stimuli (e.g. lipopolysaccharide) induced rapid coupling of S1PR1 to ß-arrestin and GFP expression in endothelium in an S1P-dependent manner (*Kono et al., 2014*). These data suggest that the S1PR1-GS mouse is a valid model to study GPCR activation in vascular ECs in vivo.

To gain insights into the molecular mechanisms of S1PR1 regulation of endothelial transcription and the heterogenous nature of S1PR1 signaling in vivo, we performed bulk transcriptome and open chromatin profiling of GFP$^{high}$ and GFP$^{low}$ aortic ECs from S1PR1-GS mice. We also performed transcriptome and open chromatin profiling of aortic ECs in which *S1pr1* was genetically ablated (*S1pr1* ECKO) (*Galvani et al., 2015*). In addition, we conducted single-cell (sc) RNA-seq of GFP$^{low}$ and GFP$^{high}$ aortic ECs. Our results show that S1PR1 suppresses the expression of inflammation-related mRNAs by inhibiting the NFκB pathway. Second, the high S1PR1 signaling ECs (GFP$^{high}$ cells) are more similar to *S1pr1* ECKO ECs at the level of the transcriptome. Third, scRNA-seq revealed eight distinct aorta-associated EC populations including six arterial EC subtypes, adventitial lymphatic ECs, and venous ECs, the latter likely from the vasa vasorum. S1PR1 signaling was highly heterogenous within these EC subtypes but was most frequent in adventitial LECs and two arterial EC populations. Immunohistochemical analyses revealed spatio-temporal regulation of aortic EC heterogeneity. In lymphatic ECs of the aorta, S1PR1 signaling restrains inflammatory and immune-related transcripts. These studies provide a comprehensive resource of transcriptional signatures in

aortic ECs, which will be useful to further investigate the multiple roles of S1P in vascular physiology and disease.

## Results

### Profiling the transcriptome of GFP[high] and GFP[low] mouse aortic endothelium

To examine S1PR1/ß-arrestin coupling in the aorta, we used the previously described S1PR1-GS (*Kono et al., 2014*) mouse strain. Mice heterozygous for the knock-in allele (*S1pr1*[ki/+]) are born at the expected Mendelian frequency (*Figure 1—figure supplement 1A*) and do not show phenotypic abnormalities (*Figure 1—figure supplement 1B and C*). However, homozygous mice (*S1pr1*[ki/ki]) showed an ~2 fold reduction in circulating lymphocytes and ~2 fold increase in lung vascular leakage of Evans Blue dye suggesting that hypomorphism of the fusion *S1pr1* allele in the signaling mouse. Therefore, all subsequent experiments were performed using heterozygous *S1pr1*[ki/+] mice harboring one allele of the *H2B-GFP* (*Tumbar et al., 2004*) reporter gene, which do not exhibit *S1pr1* hypomorphic phenotypes.

S1PR1 expression in aortic endothelium is relatively uniform (*Galvani et al., 2015*). However, S1PR1 coupling to ß-arrestin, as reported by H2B-GFP expression in S1PR1-GS mice, exhibits clear differences in specific areas of the aorta. For example, thoracic aortae of S1PR1-GS mice show high levels of GFP expression in ECs at intercostal branch points (*Galvani et al., 2015*) but not in ECs of control (*S1pr1*[+/+]) mice harboring only the *H2B-GFP* reporter allele (*Figure 1A*), confirming that GFP expression in aortic ECs is dependent on the *S1pr1* knock-in allele. The first 2–3 rows of cells around the circumference of branch point orifices exhibit the greatest GFP expression (*Figure 1A*). In addition, heterogeneously dispersed non-branch point GFP+ ECs were also observed, including at the lesser curvature of the aortic arch (*Figure 1A*). Areas of the aorta that are distal (>~10 cells) from branch points, as well as the greater curvature, exhibit relatively low frequencies of GFP+ ECs (*Figure 1A*). GFP+ mouse aortic ECs (MAECs) are not co-localized with Ki-67, a marker of proliferation, suggesting that these cells are not actively cycling (*Figure 1A*). However, fibrinogen staining was frequently co-localized with GFP+ MAECs, suggesting that ß-arrestin recruitment to S1PR1 was associated with increased vascular leak (*Figure 1A*). These findings suggest sharp spatial differences in S1PR1 signaling throughout the normal mouse aortic endothelium.

For insight into the aortic endothelial transcriptomic signature associated with high levels of S1PR1/ß-arrestin coupling, we harvested RNA from fluorescent-activated cell sorted (FACS) GFP[high] and GFP[low] MAECs and performed RNA-seq (*Figure 1B*). To identify genes that are regulated by S1PR1 signaling, we sorted MAECs from tamoxifen-treated *Cdh5*-Cre[ERT2] *S1pr1*[f/f] (*S1pr1* ECKO) and *S1pr1*[f/f] (*S1pr1* WT) littermates (*Figure 1—figure supplement 2A*). As expected, GFP[high] MAECs showed an ~20 fold increase in *eGFP* transcripts relative to GFP[low] MAECs (*Figure 1—figure supplement 2B*). We noted that GFP[high], GFP[low], *S1pr1* WT and *S1pr1* ECKO MAECs each expressed endothelial lineage genes (*Pecam1, Cdh5*) and lacked hematopoietic and VSMC markers (*Ptprc, Gata1,* and *Myocd*), validating our MAEC isolation procedure (*Figure 1—figure supplement 2C*). Efficient CRE-mediated recombination of *S1pr1* was confirmed in sorted MAECs from *S1pr1* ECKO mice (*Figure 1—figure supplement 2C*).

Differential expression analysis identified 1,103 GFP[high]-enriched and 1,042 GFP[low]-enriched transcripts (p-value<0.05) (*Figure 1C* and *Figure 1—figure supplement 2D*; see also *Supplementary file 1*). In contrast, *S1pr1* ECKO MAECs showed fewer differentially expressed genes (DEGs), with 258 up- and 107 down-regulated transcripts (*Figure 1C* and *Figure 1—figure supplement 2E*; see also *Supplementary file 1*). Intersection of these two sets of DEGs showed that only 9.5% (204 transcripts) were common (*Figure 1C* and *Supplementary file 1*), suggesting that the majority (~90%) of transcripts that are differentially expressed in MAECs from S1PR1-GS mice are not regulated by S1PR1 signaling. Rather, S1PR1/ß-arrestin coupling correlates with heterogenous EC subtypes in the mouse aorta.

Among the 204 common DEGs, 151 were both *S1pr1* ECKO up-regulated and enriched in the GFP[high] population (*Figure 1C*). In contrast, much lower numbers of transcripts were found in the intersection of GFP[high] and *S1pr1* ECKO down-regulated (seven transcripts), GFP[low] and *S1pr1* ECKO up-regulated (eight transcripts) and GFP[low] and *S1pr1* ECKO down-regulated (38 transcripts)

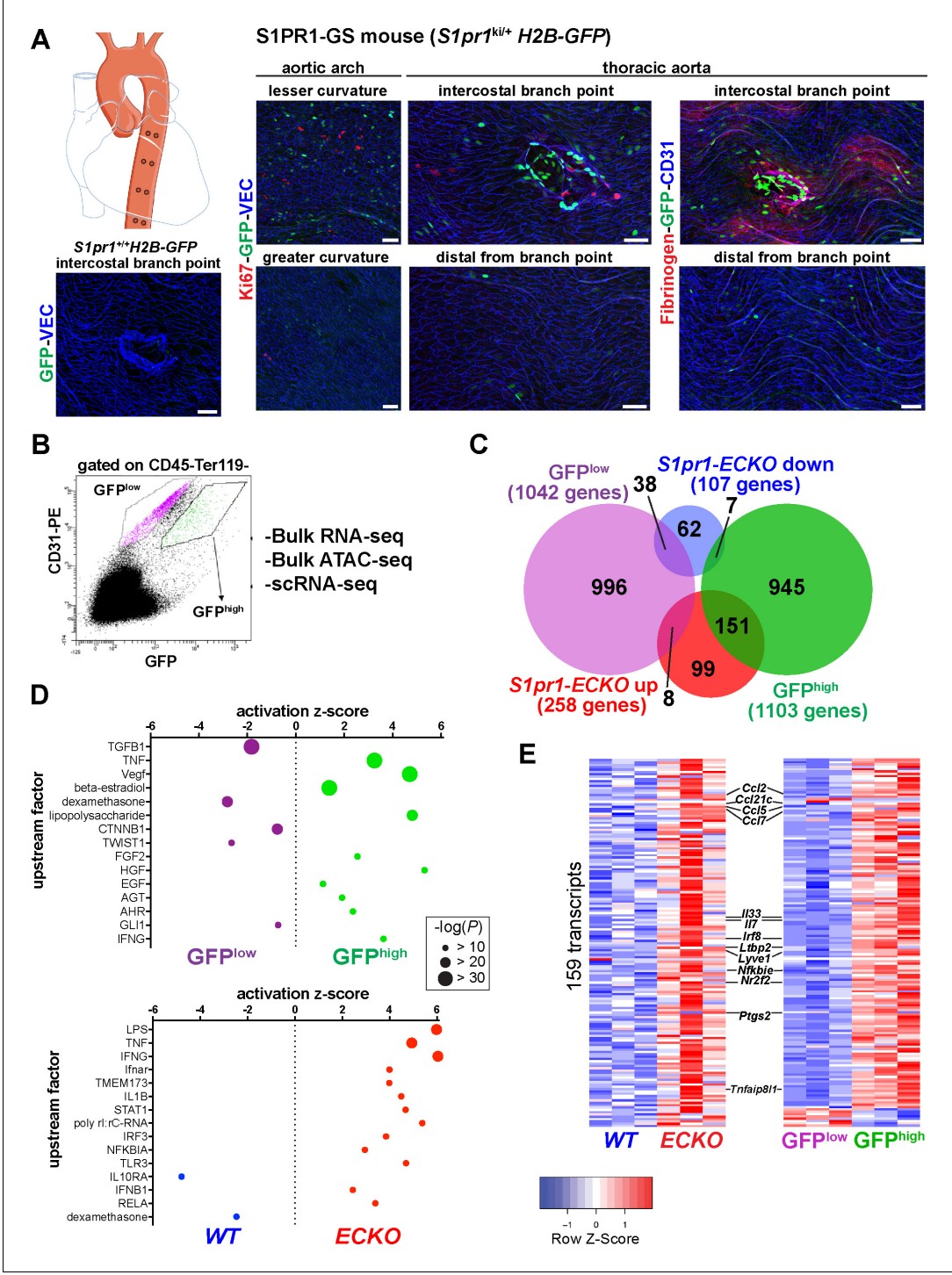

**Figure 1.** High S1PR1/ß-arrestin coupling in normal mouse aortic endothelium exhibits transcriptomic concordance with S1PR1 loss-of-function. (A) *H2B-GFP* control and S1PR1-GS mouse thoracic aorta whole-mount *en face* preparations. Representative images from different regions of the aorta are presented and show H2B-GFP (GFP), VE-Cadherin (VEC) or CD31, Ki67 (N = 3) or Fibrinogen (N = 2) immunostaining. Scale bars are 50 µM. (B) FACS gating scheme used for isolation of GFP<sup>high</sup> and GFP<sup>low</sup> MAECs showing the uncompensated CD31-PE and GFP channels. *S1pr1* ECKO and WT MAECs were isolated using the GFP<sup>low</sup> gate of this scheme (see **Figure 1— figure supplement 1A**). (C) Venn diagram showing differentially expressed transcripts in the GFP<sup>high</sup> vs. GFP<sup>low</sup> and *S1pr1* ECKO vs. WT MAECs comparisons. The number of transcripts individually or co-enriched (p-value<0.05) are indicated for each overlap (see **Supplementary file 1**). (D) Selected upstream factors identified by IPA analysis

*Figure 1 continued*

of GFP^high vs. GFP^low (top) and *S1pr1* ECKO vs. WT (bottom) MAEC comparisons. Activation Z-scores and P-values are indicated for each selected factor (see *Supplementary file 2*). (E) Expression heatmaps (row Z-scores) of the 159 *S1pr1* ECKO up-regulated transcripts also differentially expressed between GFP^high and GFP^low MAECs. Values represent individual replicates from comparison of *S1pr1* ECKO vs WT (left) and GFP^high vs GFP^low (right) and selected transcripts are labeled. For both RNA-seq experiments, three cohorts of mice were used for MAEC isolation and downstream statistical analyses (N = 3 for each experiment).

The online version of this article includes the following figure supplement(s) for figure 1:

**Figure supplement 1.** S1PR1-GS mice with a single *S1pr1*-knockin allele are phenotypically normal.

**Figure supplement 2.** RNA-seq quality control and differential gene expression between GFP^high and GFP^low and *S1pr1* ECKO and WT MAECs.

(*Figure 1C*) (detailed gene set overlaps are provided in *Supplementary file 1*). We computed the statistical significance of these gene set overlaps using the GeneOverlap R package (*Shen et al., 2013*). The GFP^high:*S1pr1* ECKO-up overlap was significant (151 transcripts, p-value=3.80E-126), as was the GFP^low:*S1pr1* ECKO-down overlap (38 transcripts, p-value=1.3E-22), and the other two tested overlaps were not significant (p-value>0.3) (*Figure 1—figure supplement 2F*). These data suggest that the transcriptome of MAECs exhibiting high S1PR1/ß-arrestin coupling (GFP^high) is more similar to that of *S1pr1* ECKO.

We used Ingenuity Pathway Analysis (IPA, Qiagen) to examine biological processes regulated by S1PR1 signaling and loss of function in MAECs. Transcripts involved in inflammatory processes were prominently up-regulated in both GFP^high and *S1pr1* ECKO MAECs (*Figure 1D*; see also *Supplementary file 2*). For example, positive tumor necrosis factor (TNF)-α, lipopolysaccharide, and interferon-γ signaling were observed in both *S1pr1* ECKO and GFP^high MAECs. In contrast, a negative glucocorticoid signature was observed in these cells (*Figure 1D*). Examples of differentially regulated transcripts are chemokines (*Ccl2*, *Clc5*, *Ccl7*, *Ccl21c*), cytokines (*Il33*, *Il7*), inflammatory modulators (*Irf8*, *Nfkbie*, *Tnfaip8l1*) and cyclooxygenase-2 (*Ptgs2*) (*Figure 1E* and *Figure 1—figure supplement 2D and E*). This suggests that S1PR1 suppresses inflammatory gene expression in mouse aortic endothelium. We noted that transcripts in the TGFß signaling pathway (*Thbs1*, *Smad3*, *Bmpr1a*, *Col4a4*, *Pcolce2*) were prominently down-regulated in the GFP^high population (*Figure 1D* and *Figure 1—figure supplement 2D*). Furthermore, both GFP^high and *S1pr1* ECKO MAECs were enriched with *Lyve1*, *Flt4*, and *Ccl21c* transcripts, which encode proteins with well-defined roles in lymphatic EC (LEC) differentiation and function (*Ulvmar and Mäkinen, 2016*; *Figure 1E*, *Figure 1—figure supplement 2G*). Taken together, these data suggest that S1PR1 represses expression of inflammatory genes in aortic endothelium and that GFP^high MAECs include aorta-associated LECs and are heterogeneous.

## Chromatin accessibility landscape of MAECs

We used the assay for transposase-accessible chromatin with sequencing (ATAC-seq) (*Buenrostro et al., 2013*) to identify putative *cis*-elements that regulate differential gene expression between GFP^high versus GFP^low and *S1pr1* WT versus *S1pr1* ECKO MAECs. ATAC-seq utilizes a hyper-active Tn5 transposase (*Adey et al., 2010*) that simultaneously cuts DNA and ligates adapters into sterically unhindered chromatin. This allows for amplification and sequencing of open chromatin regions containing transcriptional regulatory domains such as promoters and enhancers. After alignment, reads from three experiments were trimmed to 10 bp, centered on Tn5 cut sites, then merged. These merged reads were used as inputs to generate two peak sets (MACS2, FDR < 0.00001) of 73,492 for GFP^low MAECs and 65,694 for GFP^high MAECs (*Figure 2A*). MAECs isolated from WT and *S1pr1* ECKO mice harbored 93,859 and 76,082 peaks, respectively (*Figure 2A*). Peaks were enriched in promoter and intragenic regions (*Figure 2—figure supplement 1A*). We noted that the *Cdh5* gene exhibited numerous open chromatin peaks, while *Gata1* was inaccessible (*Figure 2—figure supplement 1B*). Furthermore, we observed a global correlation between chromatin accessibility and mRNA expression for all 20,626 annotated coding sequences (CDS's) in the NCBI RefSeq database (*Figure 2—figure supplement 2*). These data suggest that our ATAC-seq data is of sufficient quality for detailed interrogation.

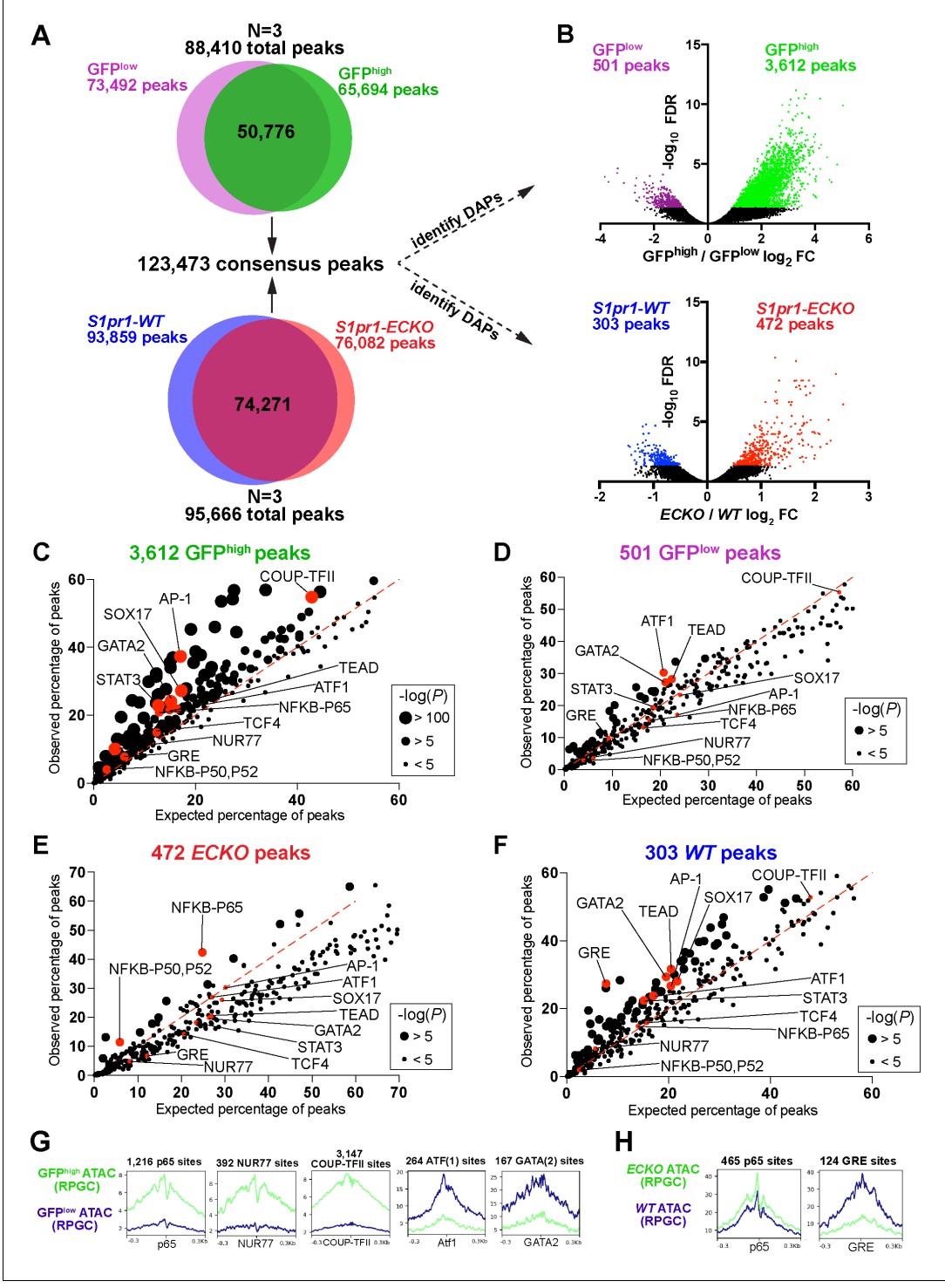

**Figure 2.** S1PR1 loss-of-function and high levels of ß-arrestin coupling are associated with an NFκB signature in open chromatin. (**A**) Venn diagrams illustrating all peaks (FDR < 0.00001) identified after analysis of individual ATAC-seq replicates of GFP$^{low}$, GFP$^{high}$, *S1pr1* ECKO, and WT MAECs, and subsequent merging of these peaks into a single consensus peak set. (**B**) Volcano plots of all peaks for both the GFP$^{high}$ vs GFP$^{low}$ and *S1pr1* ECKO vs WT MAECs. Three individual experiments were performed for GFP$^{low}$ vs GFP$^{high}$ and *S1pr1* ECKO vs WT comparisons (N = 3). Differentially accessible peaks (DAPs) were determined using edgeR (FDR < 0.05, see Materials and methods) (colored dots). (**C–F**) Transcription factor (TF) binding motif enrichment analysis of DAPs. The DAPs were input to the HOMER 'findMotifsGenome.pl' script (see also ***Supplementary file 2***) and observed vs expected frequencies of motif occurrances were plotted. (**G–H**) Graphs showing ATAC signal at predicted TF

*Figure 2 continued on next page*

*Figure 2 continued*

binding sites. ATAC-seq reads were centered on Tn5 cut sites, trimmed to 10 bp, and nucleotide-resolution bigwig files were generated using DeepTools with reads per genomic content (RPGC) normalization. Reads were subsequently centered on TF binding motifs identified in (**C–F**) and viewed as mean read densities across 600 bp windows.

The online version of this article includes the following figure supplement(s) for figure 2:

**Figure supplement 1.** ATAC-seq quality control and peak annotation.

**Figure supplement 2.** Genome-wide concordance between promoter-proximal chromatin accessibility and mRNA expression in sorted MAEC populations.

**Figure supplement 3.** Genome-wide chromatin footprinting with HINT-ATAC.

Differential chromatin accessibility analysis of GFP$^{high}$ versus GFP$^{low}$ MAECs identified 501 peaks with reduced accessibility (GFP$^{low}$ peaks) and 3612 peaks with greater accessibility (GFP$^{high}$ peaks) in GFP$^{high}$ MAECs (FDR < 0.05, *Figure 2B* and *Supplementary file 3*). For WT and *ECKO* counterparts, this analysis identified 303 peaks with reduced accessibility (*S1pr1* WT peaks) and 472 peaks with enhanced accessibility (*S1pr1* ECKO peaks) in *S1pr1* ECKO MAECs (*Figure 2B* and *Supplementary file 3*). The ~7 fold higher number of GFP$^{high}$-enriched peaks suggests that GFP$^{high}$ MAECs are more 'activated' (elevated number of chromatin remodeling events) and/or are a heterogeneous mixture of EC subtypes.

To identify relevant transcription factors (TFs), we used the Hypergeometric Optimization of Motif EnRichment (HOMER) (*Heinz et al., 2010*) suite of tools to reveal over-represented motifs in each set of differentially accessible peaks (DAPs). GFP$^{high}$ peaks were enriched with p65-NFκB, AP-1, STAT3, SOX17, COUP-TFII, and NUR77 motifs (*Figure 2C* and *Supplementary file 3*). In contrast, GFP$^{low}$ peaks showed reduced occurrence of these motifs (*Figure 2D*). *S1pr1* ECKO peaks were enriched with p65-NFκB motifs, while *S1pr1* WT peaks were markedly enriched with glucocorticoid response elements (GREs) and modestly enriched with STAT3, GATA2, ATF1, SOX17 and COUP-TFII motifs (*Figure 2E and F*). Examination of ATAC-seq reads centered on selected binding sites (p65, NUR77, COUP-TFII, ATF1, GATA2, and GRE) showed local decreases in accessibility at motif centers, suggestive of chromatin occupancy by these factors (*Figure 2G and H*).

We used the ATAC-seq footprinting software HINT-ATAC (*Li et al., 2019*) to assess genome-wide putative chromatin occupancy by TFs. HINT-ATAC identified enhanced footprinting scores at NFKB1 and NFKB2 motifs in *S1pr1* ECKO MAECs, whereas GFP$^{high}$ MAECs showed increased scores at RELA motifs and to a lesser extent at NFKB1 and NFKB2 motifs (*Figure 2—figure supplement 3A and B*). This analysis also identified motifs of the TCF/LEF family (LEF1, TCF7, TCF7L2) as GFP$^{high}$-enriched, but not *S1pr1* ECKO-enriched (*Figure 2—figure supplement 3A–D*). Consistent with HOMER analysis of DAPs, HINT-ATAC identified COUP-TFII, NUR77, TCF4, and SOX17 motifs as exhibiting greater footprinting scores in GFP$^{high}$ MAECs, while GFP$^{low}$ MAECs showed enhanced putative chromatin occupancy at ATF1 motifs.

Analysis of DAPs showed that only the p65-NFκB motif was commonly enriched between GFP$^{high}$ and *S1pr1* ECKO MAECs. This observation is consistent with our RNA-seq analysis, which identified cytokine/NFκB pathway suppression by S1PR1 signaling in MAECs (*Figure 1D*). Enrichment of COUP-TFII, NUR77, and AP-1/bZIP motifs in open chromatin of GFP$^{high}$ MAECs, but not *S1pr1* ECKO MAECs, further suggests that high levels of S1PR1/ß-arrestin coupling occurs in heterogenous populations of aortic ECs.

## Single-cell RNA-seq analysis of GFP$^{high}$ and GFP$^{low}$ MAECs reveals eight distinct EC clusters

Imaging studies demonstrated that GFP$^{high}$ MAECs are restricted to specific anatomical locations. To test the hypothesis that these represent specific EC subpopulations, we employed single-cell RNA-seq (scRNA-seq) on FACS-sorted GFP$^{high}$ and GFP$^{low}$ MAECs. In total, 1152 cells were sequenced (768 GFP$^{high}$ and 384 GFP$^{low}$) using the Smart-seq2 protocol. An average of 300,000 aligned reads/cell were obtained and corresponded to ~3200 transcripts/cell. *Cdh5* transcripts were broadly detected, consistent with endothelial enrichment of sorted cells (*Figure 3—figure supplement 1A*). *S1pr1* and *Arrb2* were also broadly detected (*Figure 3—figure supplement 1A*), suggesting that receptor activation rather than expression of these factors accounts for heterogenous

reporter expression in MAECs. *eGFP* mRNA was primarily restricted to GFP$^{high}$ MAECs, particularly in the cluster designated aEC1 (*Figure 3—figure supplement 1A*).

Analysis of GFP$^{high}$ and GFP$^{low}$ MAECs using the velocyto/pagoda2 pipeline (R code in Source Code File 3) (*Fan et al., 2016*; *La Manno et al., 2018*) revealed nine clusters upon T-distributed stochastic neighborhood embedding (t-SNE) projection (*Figure 3A*). 6 of the nine clusters grouped together in a 'cloud', whereas three clusters formed distinct populations. We used hierarchical differential expression analysis to identify signature marker genes of each cluster (*Figure 3B*).

Genes uniquely detected in one of the distinct clusters included vascular smooth muscle cell (VSMC)-specific transcripts such as *Myh11*, *Myom1*, and *Myocd* (*Figure 3B and C*; *Figure 3—figure supplement 1B*). Therefore, this cluster was designated VSMC-like as these cells may represent MAECs sorted along with fragments of VSMCs, or 'doublets' of ECs and VSMCs. Because these cells may represent contamination in an otherwise pure pool of aortic ECs, we omitted this VSMC-like cluster from subsequent analyses. The remaining eight EC clusters were further analyzed.

Lymphatic EC (LEC) markers such as *Flt4* (VEGFR3), *Prox1*, and *Lyve1*, as well as the venous marker *Nr2f2* (COUP-TFII), were detected in a distinct cluster (*Figure 3B and D*; *Figure 3—figure supplement 1B*). A smaller but nonetheless distinct cluster of ECs was also enriched with *Nr2f2* transcripts but lacked lymphatic markers, suggesting that these cells are of venous origin (*Figure 3B and E*; *Figure 3—figure supplement 1B*). Arterial lineage markers *Sox17*, *Gja5* and *Notch4* were expressed in the six clusters comprising the 'cloud' of ECs (aEC1-6) (*Figure 3B and F*; *Figure 3—figure supplement 1B*).

We individually compared LECs, vECs, and VSMCs to the remainder of ECs as a 'pseudo-bulk' cluster to generate a list of transcripts enriched (Z-score >3) for each of these three clusters. We performed the same analysis for aEC1, aEC2, aEC3, aEC4, aEC5, and aEC6, but used only arterial ECs as the comparator. For example, aEC1-enriched transcripts were identified by generating a 'pseudo-bulk' merge of all aEC2, aEC3, aEC4, aEC5, and aEC6 cells, while aEC2-enriched transcrips were compared to the pseudo-bulk merge of aEC1, aEC3, aEC4, aEC5, and aEC6. The top 32 transcripts that resulted from this analysis are shown in *Figure 3—figure supplement 2*. Among the arterial clusters, aEC1 and aEC2 harbored the greatest numbers of enriched transcripts (Z-score >3) with 411 and 1517, respectively (*Figure 3—figure supplement 3A*; see also *Supplementary file 4*). We noted that aEC5 exhibited the fewest (77) enriched transcripts (*Figure 3—figure supplement 3A*). Representative marker genes of aEC1, aEC2, aEC3, and aEC4 are shown in t-SNE embedding in *Figure 3G–J*.

LEC (97% GFP$^{high}$), vEC (100% GFP$^{high}$), aEC1 (97% GFP$^{high}$) and aEC2 (92% GFP$^{high}$) harbored the greatest proportion of GFP$^{high}$ MAECs, suggesting that S1PR1/ß-arrestin coupling is robust in these ECs subtypes (*Figure 3—figure supplement 3A and B*). In contrast, aEC3-6 contained lower frequencies of GFP$^{high}$ MAECs. aEC4 (30% GFP$^{high}$) exhibited the lowest frequency of MAECs with S1PR1/ß-arrestin coupling. The GFP$^{high}$-dominated clusters (LEC, vEC, aEC1, and aEC2) were enriched with several transcripts related to sphingolipid metabolism, such as *Spns2*, *Sptlc2*, *Ugcg*, *Enpp2*, *Ormdl3*, *Degs1*, and *Sgms2* (*Figure 3—figure supplement 4A*; see also *Supplementary file 5*). Notably, *S1pr1* transcripts were enriched in aEC1 cells by ~1.8 fold relative to the remainder of arterial ECs (*Figure 3—figure supplement 4A*; see also *Supplementary file 4*).

Pagoda2 clustering suggested that aEC1 cells were more similar to LECs and vECs than to the remainder of arterial ECs. This is illustrated by the first split of the hierarchical clustering dendrogram, which separated LEC, vEC, and aEC1 from the remainder of arterial ECs (aEC2-6) (*Figure 3B*). To identify genes that underlie the similarity between LEC, vEC, and aEC1, we identified all transcripts commonly enriched (46 transcripts, Z-score >3) and depleted (92 transcripts, Z-score < −3) in each of these three clusters when individually compared to a pseudo-bulk merge of aEC2-6 (*Supplementary file 4*). Examples of LEC, vEC, and aEC1 co-enriched transcripts were *Itga6*, *Apold1*, *Kdr*, *Fabp4*, *Robo4*, *Tcf4*, and *Adgrf5* (*Figure 3—figure supplement 4B*). Conversely, *Sod3*, *Pcolce2*, *Col4a4*, *Frzb*, *Sfrp1*, *Gxylt2*, and *Bmpr1a* were depleted from LEC, vEC, and aEC1. Notably, these depleted transcripts were highly enriched in aEC4, which is 30% GFP$^{high}$ MAECs. In contrast, LEC, vEC, and aEC1 are each >95% GFP$^{high}$.

We note that the abovementioned transcripts (*Figure 3—figure supplement 4B*) were among those most differentially expressed between GFP$^{high}$ and GFP$^{low}$ MAECs by bulk RNA-seq (*Figure 1—figure supplement 1C*). This is demonstrative of consistency between our bulk and single-cell datasets.

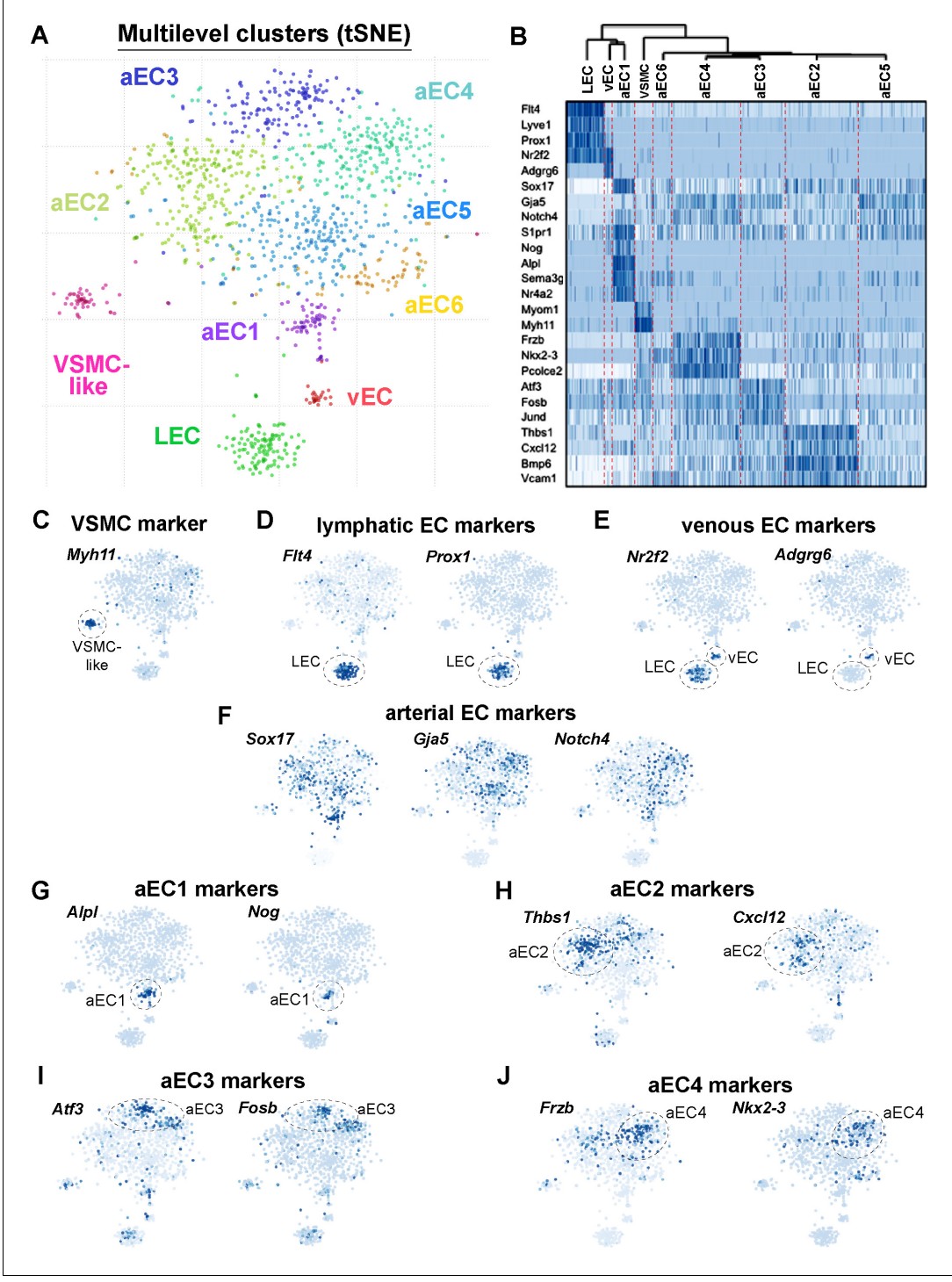

**Figure 3.** Single-cell RNA-sequencing of GFP^high and GFP^low MAECs. (**A**) t-SNE projection from Pagoda2 multilevel clustering of 767 GFP^high and 384 GFP^low MAECs. Dash-line circles highlight each of the nine clusters identified. Cells and cluster names are color-coded according to cluster assignment. (**B**) Dendrogram from hierarchical clustering and expression heatmap of selected genes. The dendrogram (top) reveals an upstream split between LEC, vEC, aEC1 and aEC2, aEC3, aEC4, aEC5, and aEC6 populations. The heatmap shows the gradient of expression, from low (white) to high (dark blue), for a selection of transcripts with distinctive expression patterns. (**C–F**) Expression of transcripts specific to VSMCs (**C**) lymphatic ECs (**D**), venous ECs (**E**) and arterial ECs (**F**) are shown on the t-SNE embedding. (**G–J**) Representative transcripts enriched in arterial EC (aEC) clusters 1 (**G**), 2 (**H**),

*Figure 3 continued on next page*

*Figure 3 continued*

3 (I) and 4 (J) are shown on the t-SNE embedding. MAECs used for scRNA-seq were isolated from two independent cohorts of S1PR1-GS mice.

The online version of this article includes the following figure supplement(s) for figure 3:

**Figure supplement 1.** Expression of cluster-defining transcripts in single GFP$^{high}$ and GFP$^{low}$ MAECs.

**Figure supplement 2.** Top marker transcripts for each of the nine clusters defined by scRNA-seq analysis of GFP$^{high}$ and GFP$^{low}$ MAECs.

**Figure supplement 3.** General features and nomenclature of the nine clusters defined by scRNA-seq analysis of GFP$^{high}$ and GFP$^{low}$ MAECs.

**Figure supplement 4.** Transcripts co-enriched in LEC, vEC, and aEC1.

---

For functional insights into arterial EC clusters, we analyzed aEC1-aEC4 enriched transcripts with the Gene Set Enrichment Analysis (GSEA) tool (*Figure 4A–D* and *Supplementary file 5*). aEC1 cells were enriched with transcripts associated with GPCR/MAPK signaling (*Rasgrp3*, *Rapgef4*, *Rgs10*, *Mapk4k3*, *S1pr1*) as well as VEGF, integrin, and tight-junction pathways (*Flt1*, *Vegfc*, *Pgf*, *Igf2*, *Vcan*, *Sema3g*, *S100a4*, *Jam2*, *Cldn5*). The aEC2 cluster presented a different profile with enriched terms related to immune/inflammatory pathways, TGFß signaling and mRNA processing. Elevated expression of *Vcam1*, *Icam1*, *Traf6*, *Cxcl12* and *NFkb1* may suggest that these ECs represent an inflammatory cluster.

In contrast, aEC3 cells were enriched with 'immediate-early' transcripts, including those of the AP-1 transcription factor family (*Atf3*, *Jun*, *Jund*, *Junb*, *Fos*, *Fosb*). Enhanced expression of *Atf3* and related TFs of the bZIP family in aEC3 may have contributed to increased chromatin accessibility at ATF, FOSB::JUN, and FOSB::JUNB binding sites in GFP$^{low}$ MAECs (*Figure 2D and G*; *Figure 2—figure supplement 3C*). Notably, a recent study of young (8 week) and aged (18 month) normal mouse aortic endothelium also identified a cluster of *Atf3*-positive cells only in young endothelium, as determined by both scRNA-seq and immunostaining for ATF3 (*McDonald et al., 2018*). Markers of these cells were also identified as top aEC3 markers (e.g. *Fosb*, *Jun*, *Jund*, *Junb*, *Dusp1*) suggesting that aEC3 cells are similar to the regenerative *Atf3*-positive cluster described by *McDonald et al. (2018)*. aEC4 cells were enriched with transcripts related to cell-ECM interations, glycosaminoglycan metabolism, and collagen formation (*Pcolce2*, *Frzb*, *Spon1*, *Col4a4*, *Mfap5*, *Hyal2*). The other two arterial clusters (aEC5 and aEC6) appeared less distinctive (i.e. harbored relatively few enriched transcripts with high Z-scores and fold change values) and therefore were not analyzed. Collectively, these data suggest that scRNA-seq identified more than four distinct MAEC subtypes with unique transcriptomic signatures.

Next, we compared our dataset with information from two recent scRNA-seq studies that also sub-categorized ECs of the normal mouse aorta (*Kalluri et al., 2019*; *Lukowski et al., 2019*). *Lukowski et al. (2019)* sequenced individual FACS-sorted Lineage$^-$CD34$^+$ cells and identified two major EC clusters, designated 'Cluster 1' and 'Cluster 2'. *Kalluri et al. (2019)* identified three major EC clusters by sequencing individual cells from whole-aorta digests. Top markers of 'Cluster 1' (*Lukowski et al., 2019*) were primarily expressed by LEC, vEC and aEC1 (*Figure 4—figure supplement 1A*). 'Cluster 1' shared markers with 'EC2' (*Kalluri et al., 2019*), such as *Rgcc*, *Rbp7*, *Cd36*, *Gpihbp1* (*Figure 4—figure supplement 1A and B*). Several 'EC2' markers, while enriched in aEC1 relative to aEC2-6 (e.g. *Flt1*, *Rgcc*), were also expressed in LEC and vEC (e.g. *Pparg*, *Cd36*, *Gpihbp1*, *Rbp7*) (*Figure 4—figure supplement 1B*). 'EC3' (*Kalluri et al., 2019*) markers were strongly enriched in LEC and vEC (e.g. *Nr2f2*, *Flt4*, *Lyve1*), and the authors noted that these cells were likely of lymphatic origin (*Figure 4—figure supplement 1B*). The remaining two clusters, 'Cluster 2' (*Lukowski et al., 2019*) and 'EC1' (*Kalluri et al., 2019*), strongly resembled the aggregate of aEC2-6 as they were enriched with *Gata6*, *Vcam1*, *Dcn*, *Mfap5*, *Sfrp1*, *Eln*, and *Cytl1* (*Figure 4—figure supplement 1A and B*). Taken together, these information from three independent groups strongly suggest that a major source of heterogeneity in the normal adult mouse aorta, as revealed by scRNA-seq analysis, includes differences between LEC, vEC, aEC1 and aEC2-6.

## Localization of heterogenous arterial EC populations

Among the arterial populations, aEC1 and aEC2 exhibited the highest frequencies of S1PR1/ß-arrestin coupling (>90%). Thus, we addressed the anatomical location of these arterial clusters in the

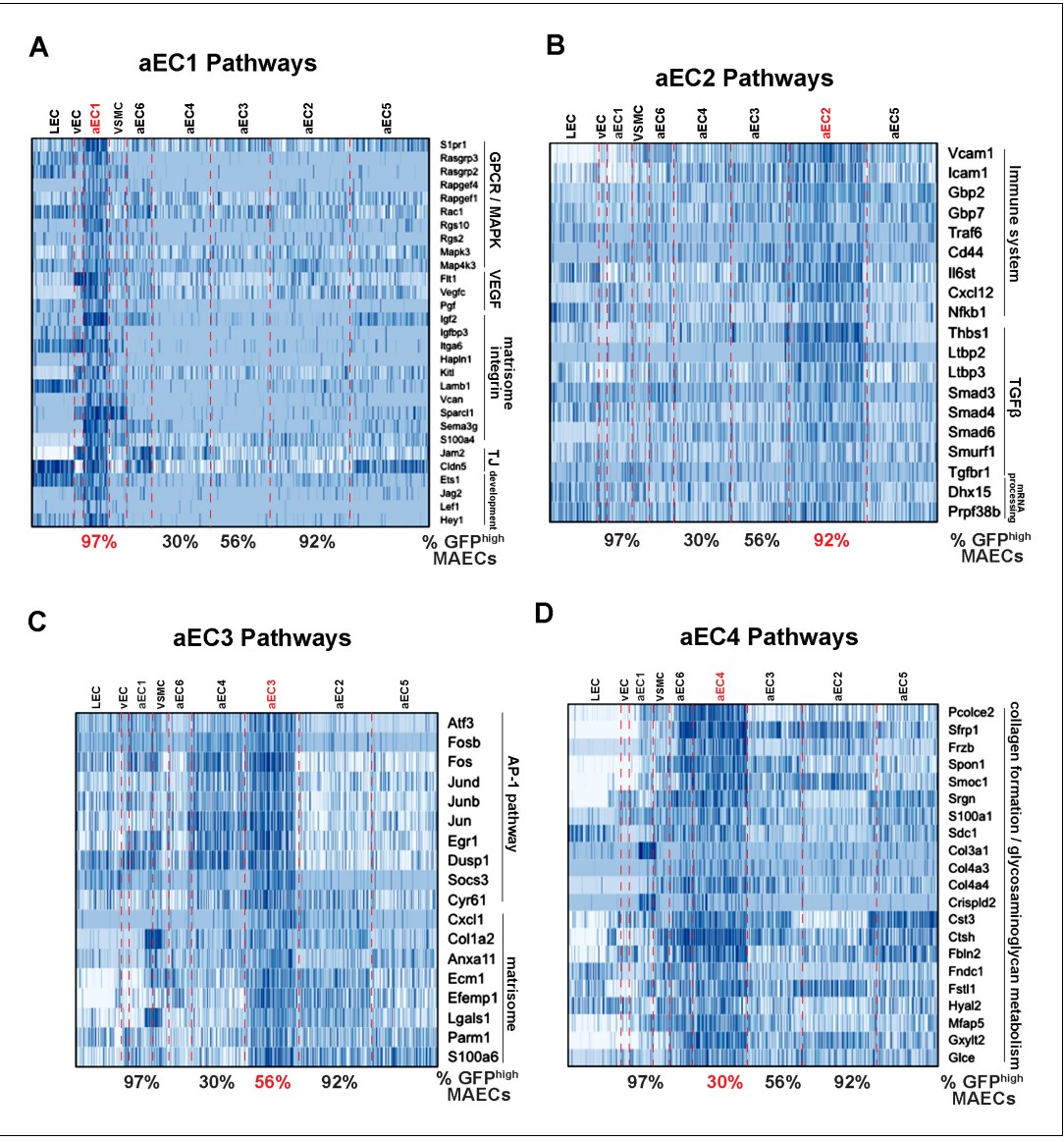

**Figure 4.** Functional annotation of arterial MAECs clusters aEC1, aEC2, aEC3, and aEC4. (A–D) Selected pathways from GSEA analysis of cluster-enriched transcripts. Pathways enriched in aEC1 (A), aEC2 (B), aEC3 (C), and aEC4 (D) are shown with representeative transcripts identified in each pathway (see *Supplementary file 5*). The percent of GFP[high] cells in each of the four analyzed clusters are indicated at the bottom each heatmap. The online version of this article includes the following figure supplement(s) for figure 4:

**Figure supplement 1.** Expression of aortic EC cluster markers from *Lukowski et al. (2019)* and *Kalluri et al. (2019)* in LEC, vEC, and aEC1-6.

normal adult mouse aorta by immunolocalization of markers. We utilized antibodies against noggin (NOG), alkaline phosphatase (ALPL), and integrin alpha-6 (ITGA6), which are highly enriched in aEC1, as well as fibroblast-specific protein-1 (FSP1, encoded by *S100a4*) and claudin-5 (CLDN5), which are enriched in, but not exclusive to, aEC1 (*Figure 5A*). We also utilized antibodies against thrombospondin-1 (TSP1, encoded by *Thbs1*) and vascular cell adhesion molecule 1 (VCAM1), which are enriched in aEC2 and depleted in aEC1 (*Figure 5A*).

*En face* immunofluorescence staining showed that NOG and ITGA6 were expressed by GFP[high] MAECs at intercostal branch points (*Figure 5B and C*). These GFP[high] MAECs comprise the first 2–3 rows of cells around the circumference of branch orifices and include ~20–30 cells in total. In

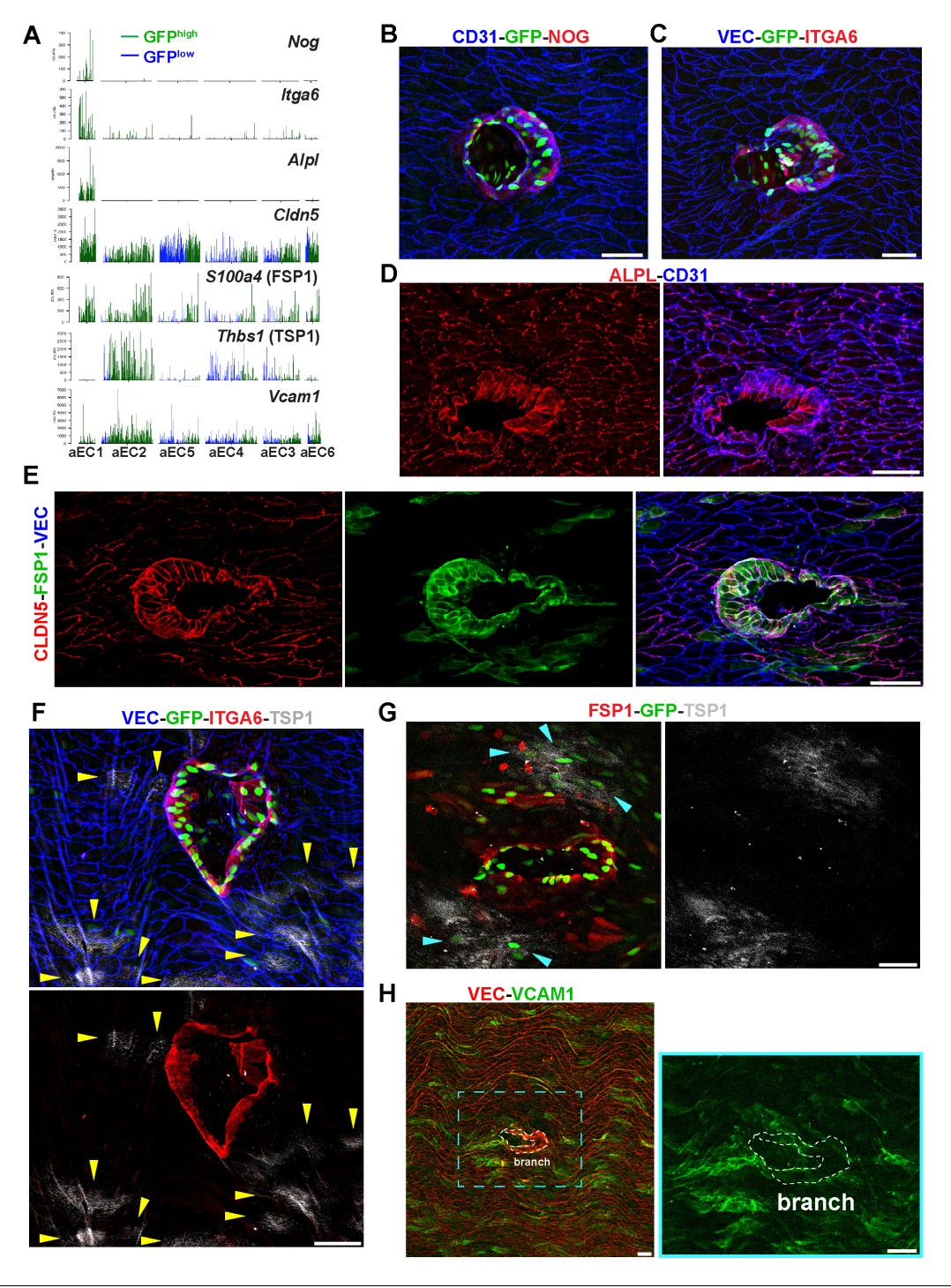

**Figure 5.** Identification of aEC1 as arterial ECs around the circumference of branch point orifices. (**A**) Barplots of scRNA-seq read counts for *Nog*, *Itga6*, *Alpl*, *Cldn5*, *S100a4*, *Thbs1*, and *Vcam1* in arterial EC clusters (aEC1-6). Each bar represents a single cell, either GFP^high (green) or GFP^low (blue). (**B–H**) Confocal images of whole-mount *en face* preparations of mouse thoracic aortae centered on intercostal branch points. Immunostaining of S1PR1-GS mouse aortae for noggin (NOG) and CD31 (**B**), or integrin alpha-6 (ITGA6) and VE-Cadherin (VEC) (**C**). Immunostaining of C57BL/6J mouse aortae for CD31 and alkaline phosphatase, tissue-nonspecific isozyme (ALPL) (**D**) or claudin-5 (CLDN5), FSP1 (S100A4), and VEC (**E**). (**F**) Immunostaining of S1PR1-GS mouse aorta for ITGA6, thrombospondin 1 (TSP1), and VEC. Yellow arrows indicate TSP1+ ECs. (**G**) Immunostaining of an S1PR1-GS mouse aorta for FSP1 and TSP1. The circumference of an intercostal branch point harbors FSP1+GFP+ cells,

*Figure 5 continued on next page*

*Figure 5 continued*

wereas TSP1+GFP+ cells (cyan arrows) are distal from the circumference of the branch point orifice. (H) Immunostaining of a mouse aorta for vascular cell adhesion molecule 1 (VCAM1) and VEC. Two rows of cells exhibiting the morphology and VEC-localization of cells around branch point orifices are outlined. Images are representative of observations from 3 (B, C, E–H) and 2 (D) mice. Scale bars are 50 μM.

The online version of this article includes the following figure supplement(s) for figure 5:

**Figure supplement 1.** Patchy TSP1 immunoreactivity is endothelial and excluded from thoracic branch point orifices.

**Figure supplement 2.** VCAM1 is LPS-inducible and expression is observed in GFP+ ECs distal from thoracic branch point orifices.

---

contrast, cells that are more than 2–3 cells away from branch point orifices did not express NOG or ITGA6 (*Figure 5B and C*). This was also seen for cytoplasmic staining of ALPL (*Figure 5D*). CLDN5 and FSP1 staining demarcate branch point MAECs but also exhibited heterogeneous staining of surrounding ECs (*Figure 5E*).

In contrast, TSP1 staining was exluded from ITGA6-expressing GFP$^{high}$ MAECs at branch points (*Figure 5F*; see also *Figure 5—figure supplement 1A*). Rather, non-branch point GFP$^{high}$ MAECs expressed TSP1 in a patchy pattern (*Figure 5G*). We confirmed the endothelial nature of TSP1 immunoreactivity by staining sagittal sections of thoracic aortae (*Figure 5—figure supplement 1B*).

Similar to TSP1, VCAM1 staining was low at branch points, while surrounding cells showed heterogeneous levels of immunoreactivity (*Figure 5H*). We identified isolated GFP+ cells and patches of GFP+ cells distal from branch points with high VCAM1 immunoreactivity (*Figure 5—figure supplement 2A*). As expected, intraperitoneal LPS administration induced VCAM1 expression uniformly in aortic ECs (*Figure 5—figure supplement 2B*). These data suggest that the aEC1 population includes MAECs that are anatomically specific to branch points, while the aEC2 population is located distally and heterogeneously from branch points but nonetheless exhibits S1PR1/ß-arrestin signaling.

## Analysis of branch point-specific arterial ECs cluster aEC1

Having identified the aEC1 cluster as including cells which demarcate thoracic branch points orifices, we sought to characterize these unique cells in further detail. As shown in *Figure 4A*, aEC1-enriched transcripts are associated with a diverse range of signaling pathways, including MAPK/GPCR, VEGF, and integrin signaling. Among aEC1-enriched transcripts, 16 were up-regulated in *S1pr1* ECKO MAECs, five were down-regulated (including *S1pr1*) and the remaining 390 were not differentially expressed (*Figure 6A*; see also *Supplementary file 4*). The 16 *ECKO* up-regulated transcripts included positive regulators of angiogenesis (*Pgf*, *Apold1*, *Itga6*, *Kdr*) (*Mirza et al., 2013*; *Olsson et al., 2006*; *Primo et al., 2010*), regulators of GPCR signaling (*Rgs2*, *Rasgrp3*), and *Cx3cl1* (Fractalkine), which encodes a potent monocyte chemoattractant (*White and Greaves, 2012*). We noted that several aEC1-enriched transcripts were also expressed in LEC and vEC, but nonetheless were depleted from the remainder of arterial ECs (aEC2-6) (*Figure 6B*). We examined the 25 transcripts most specific to aEC1 (log$_2$ [fold-change] vs. all ECs > 4) and observed that 5 of these (*Dusp26*, *Eps8l2*, *Hapln1*, *Lrmp*, and *Rasd1*) showed differential expression upon loss of S1PR1 function in MAECs (*Figure 6C*). Therefore, the majority of transcripts specific to aEC1 do not appear to require S1PR1 signaling for normal levels of expression. For example, protein levels of the aEC1 marker ITGA6 were not markedly affected at branch point orifices in *S1pr1* ECKO animals (*Figure 6D*). The ~2 fold increase in *Itga6* transcript levels in *S1pr1* ECKO MAECs may be due to expression in heterogeneous (non aEC-1) populations, reflecting increases in LECs and/or aEC2-6. Nonetheless, these data suggest that S1PR1 signaling is required for normal expression of some, but not the majority, of transcripts enriched in aEC1 cells located at orifices of intercostal branch points.

We addressed whether circulatory S1P is required for S1PR1/ß-arrestin coupling in arterial aortic endothelium by genetically deleting the two murine sphingosine kinase enzymes, *Sphk1* and *Sphk2*. We generated S1PR1-GS mice harboring *Sphk1$^{f/f}$* and *Sphk2$^{-/-}$* alleles, bred this strain with the tamoxifen-inducible *Rosa26-Cre-ER$^{T2}$* allele, and induced *Sphk1* deletion by tamoxifen injection into adult mice (see Materials and methods). 5–6 weeks after tamoxifen administration, plasma S1P

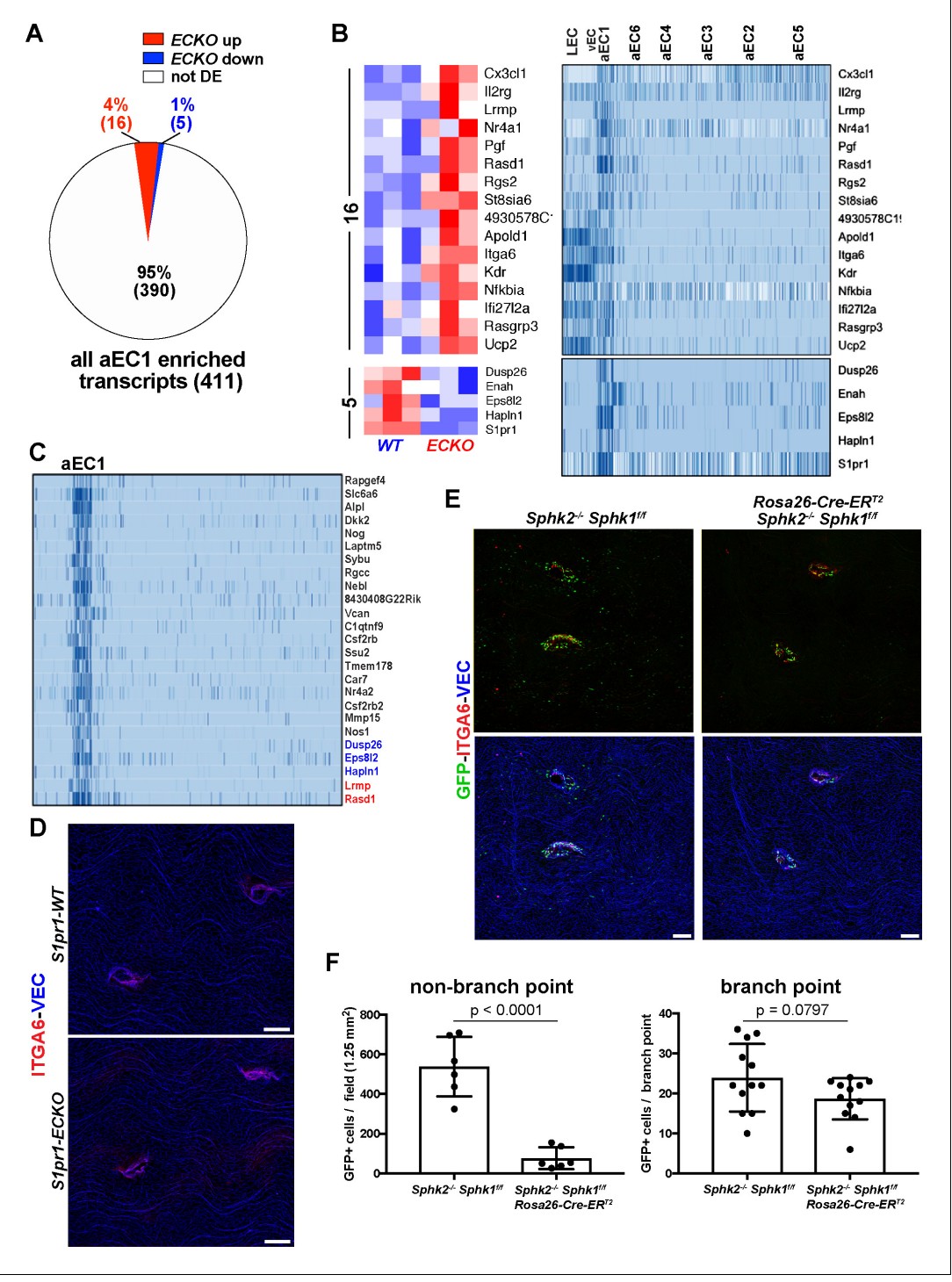

**Figure 6.** Gene expression in aEC1 cells is largely independent of S1P/S1PR1 signaling. (**A**) Pie chart of all aEC1-enriched transcripts (versus aEC2-6) indicating those which were also differentially expressed in *S1pr1* ECKO MAECs. (**B**) Heatmap (row Z-scores) of the 20 transcripts that were both S1PR1-regulated and aEC1-enriched (left). Expression of these transcripts in each cluster from scRNA-seq analysis is shown (right). (**C**) Gene expression in aEC1 was compared to all ECs (LEC, vEC, and aEC2-6 collectively). All transcripts expressed greater than 16-fold higher in aEC1 are shown (25 transcripts total). Red, blue and black transcript names indicate up-regulated, down-regulated or similar levels of expression in *S1pr1* ECKO MAECs, respectively. (**D**) Immunostaining of *S1pr1* ECKO and WT aortae whole-mount *en face* preparations for ITGA6 and VEC. Images are representative of observations from 2 pairs of animals (N = 2). (**E**) ITGA6 and VEC immunostaining of whole-mount *en face* preparations of

*Figure 6 continued*

thoracic aortae from S1PR1-GS mice bearing *Sphk2*$^{-/-}$ *Sphk1*$^{f/f}$ or *Sphk2*$^{-/-}$*Sphk1* $^{f/f}$*Rosa26-Cre-ER*$^{T2}$ alleles. (F) Quantification and statistical analyses (unpaired t-test) of GFP+ EC at branch point (12 branches) and non-branch point (six fields) locations from 2 pairs of mice (N = 2). Quantified fields are as shown in (E). Branch point EC were defined as cells included in the first three rows around the edge of orifices. Only GFP+ EC in the same Z-plane as surrounding arterial EC were counted (GFP+ EC of intercostal arteries were in a different Z-plane and therefore were not counted as branch point EC). Scale bars are 100 μM.

concentrations in Cre- animals were 631 ± 280 nM, whereas S1P was undetectable in plasma from Cre+ mice. Cre+ mice had ~7 fold fewer non-branch point GFP+ ECs (*Figure 6E and F*), suggesting that S1PR1/ß-arrestin coupling in aEC2 cells is ligand-dependent. In contrast, the number of branch point GFP+ EC was not significantly different between Cre+ and Cre- animals, suggesting ligand-independent S1PR1/ß-arrestin coupling in aEC1 population (*Figure 6E and F*). Furthermore, ITGA6 expression at branch point orifices was unaltered in Cre+ mice and therefore is independent of circulatory S1P. Taken together, these data suggest that the unique transcriptome of aEC1 cells is largely independent of S1P/S1PR1 signaling while in aEC2 (and perhaps others), ligand-dependent S1PR1 signaling predominates.

## Spatio-temporal and molecular differences between branch point-specific arterial EC cluster aEC1 and non-branch point ECs

For insight into transcriptional regulatory mechanisms in aEC1 cells, we identified all TFs enriched and depleted in this cluster (*Figure 7—figure supplement 1A*; see also *Supplementary file 5*). Among arterial ECs, TFs highly enriched (Z-score >7) in aEC1 are *Hey1*, *Nr4a2* (NURR1), *Nr4a1* (NUR77), *Sox17*, *Ebf1*, and *Bcl6b* (*Figure 7A* and *Figure 7—figure supplement 1A and B*). *Lef1* transcripts were detected at significant levels only in aEC1 cells (*Figure 7—figure supplement 1A and B*). Transcripts encoding other TFs, such as *Tcf4*, *Ets1*, *Sox18*, *Epas1*, *Mef2c*, and *Tox2*, were aEC1-enriched (Z-score >3 and<7) but more heterogeneously distributed in other arterial clusters (*Figure 7A* and *Figure 7—figure supplement 1A*). These data are consistent with our ATAC-seq analysis, which showed over-representation of SOX17, TCF4, and NUR77 motifs in chromatin specifically open in GFP$^{high}$ MAECs. In contrast, *Gata6* (ubiquitous among aEC2-6) and *Gata3* (heterogeneous among aEC2-6) were both depleted from aEC1 (*Figure 7A* and *Figure 7—figure supplement 1A and B*).

Immunostaining of thoracic aorta *en face* preparations for LEF1 showed nuclear immunoreactivity in GFP$^{high}$ ECs at branch point orifices but not in adjacent ECs (*Figure 7B*). We noted that all LEF1+ cells also exhibited ITGA6 expression, confirming these two proteins as markers of aEC1 cells at branch point orifices (*Figure 7B*). These data suggest involvement of LEF1, a downstream TF of Wnt/ß-catenin signaling, in regulating gene expression in aEC1 cells.

For broader insight into TF activity near aEC1 genes, we extracted all GFP$^{high}$ and GFP$^{low}$ merged peaks that intersected a 100 kb window centered on the TSSs of all aEC1-enriched (Z-score >3) and -depleted (Z-score < −3) genes. HOMER analysis of these peaks showed enrichment of SOX17, 'Ets1-distal', MEF2C, and NUR77 motifs near the aEC1-enriched genes (*Figure 7—figure supplement 1C*). In contrast, GATA motifs (GATA2, GATA3, GATA6), as well as the NKX2.2 motif, were enriched near aEC1-depleted genes (*Figure 7—figure supplement 1C*). Collectively, these data suggest roles for specific transcription factors, such as LEF1, SOX17, NUR77, and GATA6, in mediating transcriptional events which distinguish aEC1 from the remainder of aortic arterial ECs.

Examination of postnatal day 6 (P6) S1PR1-GS aortae showed that ECs at branch point orifices expressed GFP and ITGA6 in a manner similar to adult S1PR1-GS mice (*Figure 7C*). We noted that non-branch point GFP+ EC were less frequent in P6 mice relative to young adults (*Figure 7C*). Quantification of non-branch point GFP+ EC in aortae of P6, P12, young adult (3–4 months), and 15 month old mice revealed a ~ 5 fold increase from P12 to young adult (*Figure 7D* and *Figure 7—figure supplement 2A–D*). There was no appreciable difference in non-branch point GFP+ EC frequency between P6 and P12 or between young adult and 15 months. In contrast, the number of GFP+ EC at branch points was stable from P12 throughout adulthood (*Figure 7—figure supplement 2E*). These data suggest that S1PR1/ß-arrestin coupling in non-branch point ECs increases with post-natal age.

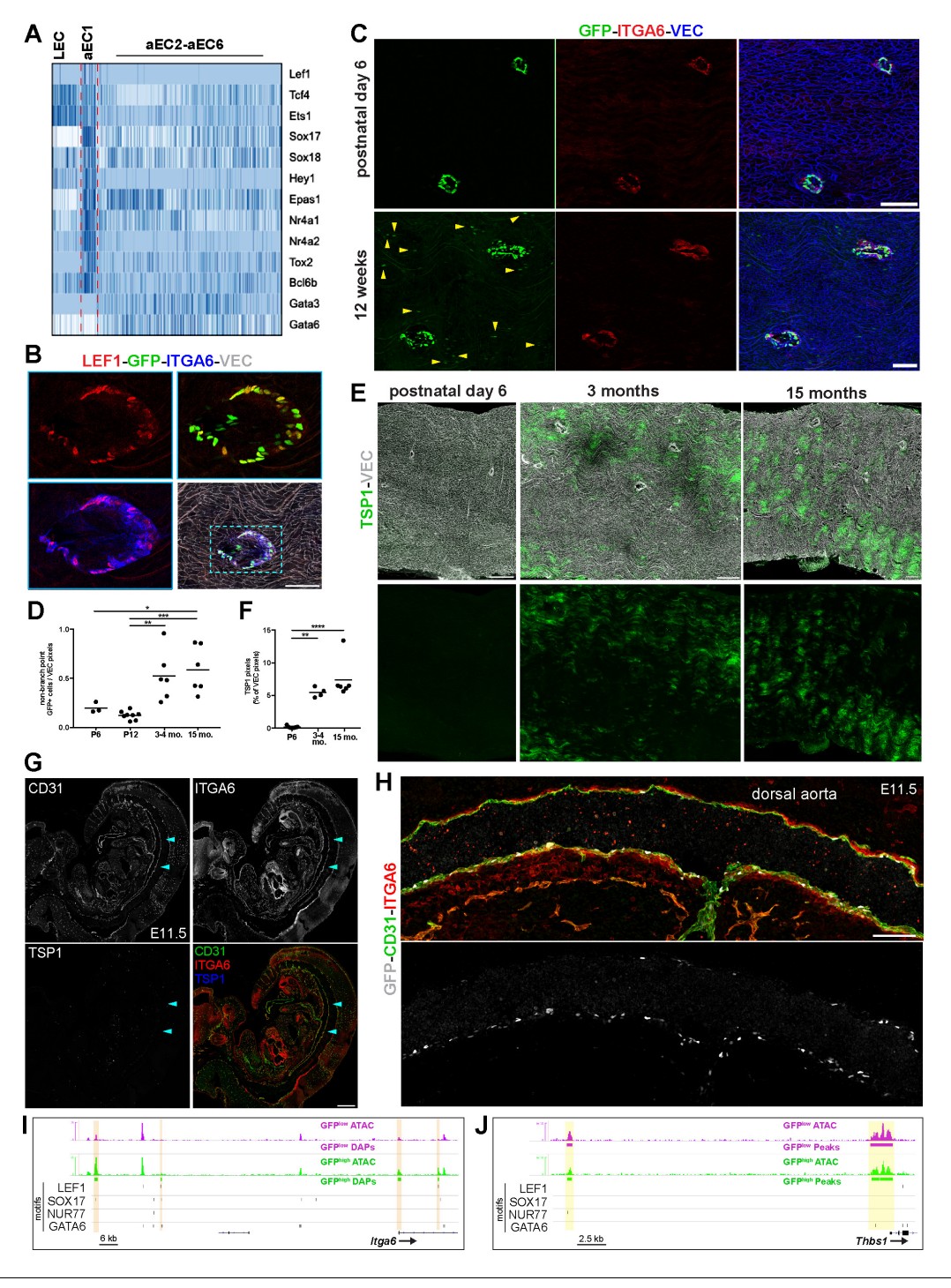

**Figure 7.** Spatio-temporal and molecular differences between branch point-specific arterial EC cluster aEC1 and non-branch point ECs. (**A**) Heatmap of selected transcription factors enriched and depleted in aEC1 cells (see also *Figure 7—figure supplement 1A*). (**B**) Immunostaining of an S1PR1-GS mouse thoracic aorta for lymphoid enhancer-binding factor 1 (LEF1), ITGA6, and VEC. Image is representative of observations from two mice. Scale bar is 100 µM. (**C**) Immunostaining of postnatal day 6 (**P6**) (n = 2) and 12 week (n = 3) S1PR1-GS mice thoracic aortae for ITGA6 and VEC. Arrows indicate non-branch point GFP+ EC. Scale bars are 100 µM. (**D**) Quantification of non-branch point GFP+ EC in thoracic aorta *en face* preparations from P6 (n = 2), P12 (n = 3), 3–4 month-old (n = 3), and 15 month old (n = 2) mice. Each dot is a field. Representative fields are shown in *Figure 7—figure supplement 2*. Non-branch point GFP pixels were normalized to total VEC pixels per field. One-way ANOVA

*Figure 7 continued on next page*

*Figure 7 continued*

followed by Bonferroni's *post hoc* test, $^*P = 0.04$; $^{**}P = 0.04$; $^{***}P = 0.0008$. (**E**) Representative images of TSP1 and VEC immunostaining of thoracic aorta *en face* preparations from P6, 3 month old, 15 month old mice. Scale bars are 200 µM. (**F**) Quantification of TSP1 pixels normalized to total VEC pixels per field from P6 (n = 3), 3 month old (n = 2), and 15 month old (n = 2) mice. One-way ANOVA followed by Bonferroni's *post hoc* test, $^{**}P = 0.001$; $^{****}P < 0.0001$. (**G**) Sagittal cryosection (14 µM) of an E11.5 S1PR1-GS mouse embryo immunostained for CD31, ITGA6, and TSP1. Arrows indicate the dorsal aorta. Scale bar is 500 µM. (**H**) The dorsal aorta at higher magnification with CD31, ITGA6, and GFP channels shown. Scale bar is 100 µM. (**G**) and (**H**) are representative of observations from two embryos. (**I–J**) Genome browser image of GFP^high^ and GFP^low^ MAECs ATAC-seq signal (RPGC normalized) at *Itga6* and *Thbs1* (TSP1) loci. Peaks with increased accessibility in GFP^high^ MAECs (GFP^high^ DAPs) are indicated in (**I**). All GFP^high^ and GFP^low^ MAECs peaks are shown in (**J**). All LEF1, nuclear receptor subfamily four group A member 1 (NUR77), transcription factor SOX-17 (SOX17), and transcription factor GATA-6 (GATA6) motifs identified in consensus peaks (*Figure 2A*) are also shown. Orange bars in (**I**) highlight GFP^high^ MAECs DAPs containing LEF1, SOX17, and/or NUR77 motifs. Yellow bars in (**J**) highlight peaks containing GATA6 motifs. The online version of this article includes the following figure supplement(s) for figure 7:

**Figure supplement 1.** Transcription factors enriched and depleted in aEC1 cells.
**Figure supplement 2.** Early postnatal, young adult, and aged S1PR1-GS mouse thoracic aortae.
**Figure supplement 3.** Embryonic expression of ITGA6, TSP1, and aEC1-enriched and –depleted transcripts.
**Figure supplement 4.** Identification of embryonic lymphatic ECs with S1PR1/ß-arrestin coupling.
**Figure supplement 5.** Differential chromatin accessibility near aEC1- and aEC4-enriched genes is coincident with specific DNA motifs.

To test the hypothesis that aEC2-like (GFP^high^) cells occur at greater frequency during aging, aortae from P6, young adult, and 15 month old mice were immunostained for VEC and TSP1, which is abundantly expressed by aEC2 cells. TSP1 immunoreactivity was markedly reduced to near-undetectable levels in P6 mice relative to older counterparts (*Figure 7E and F*). Taken together, these data suggest that TSP1-expressing (aEC2-like) cells increase in frequency while aEC3-like cells disappear over time in the aorta intima, which was described previously (*McDonald et al., 2018*).

Next, we explored the expression of *Itga6* and *Thbs1* (TSP1) in mouse embryos. First, we examined the 'mouse organogenesis cell atlas' database (https://oncoscape.v3.sttrcancer.org/atlas.gs.washington.edu.mouse.rna/landing), which includes scRNA-seq data from E9.5 to E13.5 mouse embryos (*Cao et al., 2019*). *Itga6* transcripts were in many cell types, but were most abundant in endothelial cells (*Figure 7—figure supplement 3A*). In contrast, high *Thbs1* expression was restricted to megakaryocytes and endothelial expression was ~40 fold lower than endothelial *Itga6* expression. Consistently, E11.5 S1PR1-GS mouse embryo sections immunostained for TSP1, ITGA6, and CD31 showed widespread endothelial ITGA6 immunoreactivity and no endothelial TSP1 immunoreactivity (*Figure 7G*). The dorsal aorta was also ITGA6+ (*Figure 7G and H*), while TSP1+ cells with hematopoietic morphology were present in the lung bud and hepatic primordium (*Figure 7—figure supplement 3B*). Consistent with previous observations (*Kono et al., 2014*), the dorsal aorta harbored GFP+ ECs (*Figure 7H* and *Figure 7—figure supplement 3C*). Images taken at a higher resolution confirmed the endothelial expression of ITGA6 in both GFP+ and GFP- cells in the dorsal aorta (*Figure 7—figure supplement 3D*). We also observed CD31+ITGA6+LYVE1+GFP+ cells in the cardinal vein region, suggesting that S1PR1/ß-arrestin coupling occurs in lymphatic ECs during developmental lymphangiogenesis (*Figure 7—figure supplement 4A and B*).

We sought to determine if aEC1-enriched transcripts other than *Itga6* and *Thbs1* show evidence of temporally-regulated expression in aortic endothelium. Examination of the embryonic gene expression database (*Cao et al., 2019*) revealed that other aEC1-enriched transcripts, such as *Igf2*, *Flt1*, *Slc6a6*, and *Lef1* were expressed at greater levels in embryonic ECs relative to aEC1-depleted transcripts (e.g. *Vcam1*, *Frzb*, *Pcolce2*, *Cyp1b1*, *Sod3*) (*Figure 7—figure supplement 3D*) that are enriched in aEC2 and/or aEC4. Thus, aortic endothelium exhibits spatio-temporal regulation of genes such as *Itga6* and *Thbs1*, with expression of aEC1 genes such as *Itga6* becoming restricted to branch point orifices over time while expression of aEC2-enriched genes (e.g. *Thbs1*) increases throughout the aorta intima after P6.

We further explored potential roles for LEF1, SOX17, NUR77, and GATA6 in aEC1 gene expression by examining binding sites for these factors near genes encoding aEC1-enriched transcripts. A prominent GFP^high^-specific peak in the first intron of *Itga6* harbored a LEF1 motif (*Figure 7I*). In

addition, a GFP^high^-enriched peak ~115 kb upstream of the transcription start site (TSS) harbored a SOX17 motif. The *Nog* gene contained SOX17 and NUR77 motifs in upstream GFP^high^-specific peaks (*Figure 7—figure supplement 4A*). Each of these putative enhancers lacked GATA6 motifs. In contrast, open chromatin near the *Thbs1* (TSP1) gene lacked SOX17 and LEF1 sites and instead contained three GATA6 sites (*Figure 7J*). Furthermore, the *Frzb* and *Pcolce2* genes, which encode aEC4-enriched transcripts and were up-regulated in GFP^low^ MAECs (*Figure 1—figure supplement 2D*), harbored intronic GFP^low^ MAEC-specific peaks containing GATA6 binding sites (*Figure 7—figure supplement 4B*).

Together, these findings characterize the aortic branch point-specific arterial EC subpopulation designated aEC1. The data strongly suggest that the GFP^high^ status of MAECs at branch point orifices, as well as their unique gene expression program, is not dynamic throughout postnatal life. Rather, the gene expression specification of these cells likely occurs during development in tandem with epigenetic changes (i.e. chromatin accessibility) and is stable throughout adulthood. These cells have a unique anatomical location in postnatal mice and exhibit high S1PR1/ß-arrestin coupling. The transcriptome of this EC subpopulation does not appear to be directly regulated by S1PR1 signaling. Rather, a combination of TFs, such as NUR77, NURR1, SOX17, HEY1, and LEF1, likely regulate cluster-defining transcripts in these cells.

## S1PR1 signaling in adventital lymphatic ECs regulates immune and inflammatory gene expression

To locate the aorta-associated LEC with a high frequency (97%) of S1PR1/ß-arrestin coupling, we utilized antibodies against LYVE1 and VEGFR3 (*Flt4*). LYVE1 marks most but not all LEC subtypes and VEGFR3 is a pan-LEC marker (*Wang et al., 2017*). Sagittal sections of the S1PR1-GS mouse thoracic aorta revealed that a subset of adventitial LYVE1+ LECs are GFP+ and thus exhibit S1PR1/ß-arrestin coupling (*Figure 8A*).

Next, we prepared whole-mounts of thoracic aortae and collected confocal Z-stacks of only the adventitial layer (*Figure 8B* and *Figure 8—figure supplement 1B*). We observed three distinct expression patterns: VEGFR3+LYVE1+GFP+ (orange stars), VEGFR+LYVE1^low^GFP+ (cyan stars), and VEGFR3+LYVE1+GFP- (magenta stars). We noted that VEGFR3+LYVE1+GFP- areas were associated with blind-ended bulbous structures that bear resemblance to LYVE1+ lymphatic capillaries (*Ulvmar and Mäkinen, 2016*). In contrast, VEGFR+LYVE1^low^GFP+ structures were morphologically similar to collecting lymphatic vessels, which are also LYVE1^low^ in the mouse ear dermis (*Ulvmar and Mäkinen, 2016*).

In addition to expression in aEC1, *Itga6* transcripts were detected in vEC and LEC populations (*Figure 8—figure supplement 1A*). Consistently, we observed GFP+ITGA6+LYVE1+ LEC on the adventitial side of the aorta, in proximity to a GFP+ITGA6+LYVE1- arterial branch point (*Figure 8—figure supplement 1C*).

We examined the role of LEC-derived S1P in LEC S1PR1/ß-arrestin coupling by generating S1PR1-GS mice deficient in lymph S1P (*S1pr1*^ki/+^:*Sphk1*^f/f^:*Sphk2*^-/-^:*Lyve1-Cre*^+^) (*Pham et al., 2010*). LECs were identified in mesenteric vessels by immunostaining for the LEC-specific transcription factor prospero homeobox protein 1 (PROX1). Collecting lymphatics, which exhibit coverage of α-smooth muscle actin (ASMA) positive cells (*Wang et al., 2017*), were identified by ASMA co-staining. *H2B-GFP* control mice showed low levels of GFP expression in lymphatic valves and the remainder of LECs were GFP- (*Figure 8C*). In stark contrast, GFP+ LECs were observed at high frequency (73%) in ASMA+ structures but not in ASMA- structures (6%) of S1PR1-GS mice (*Figure 8D and E*). This suggests that S1PR1/ß-arrestin coupling occurs primarily in collecting lymphatics. Mice lacking S1P in lymph (*Lyve1-Cre*^+^) exhibited a 10-fold reduction in GFP+ LEC in ASMA+ structures (7%), indicating that the ligand S1P induces S1PR1/ß-arrestin coupling in collecting lymphatic vessels. These data are consistent with abundant LEC expression of the S1P transporter *Spns2* (*Figure 3—figure supplement 4A*), which is also required for normal levels of lymph S1P (*Simmons et al., 2019*). Taken together, scRNA-seq analysis of aortic ECs identified two anatomically distinct arterial EC populations (branch point and non-branch point), as well as a collecting lymphatic-like adventitial LEC population, each of which shows high S1PR1/ß-arrestin coupling.

For insight into S1PR1-mediated gene expression in aorta-associated LECs, we divided *S1pr1* ECKO up-regulated genes (*Figure 1C* and *Figure 1—figure supplement 2E*) according to their cluster assignments from scRNA-seq analysis. 30% of up-regulated genes were enriched in the LEC

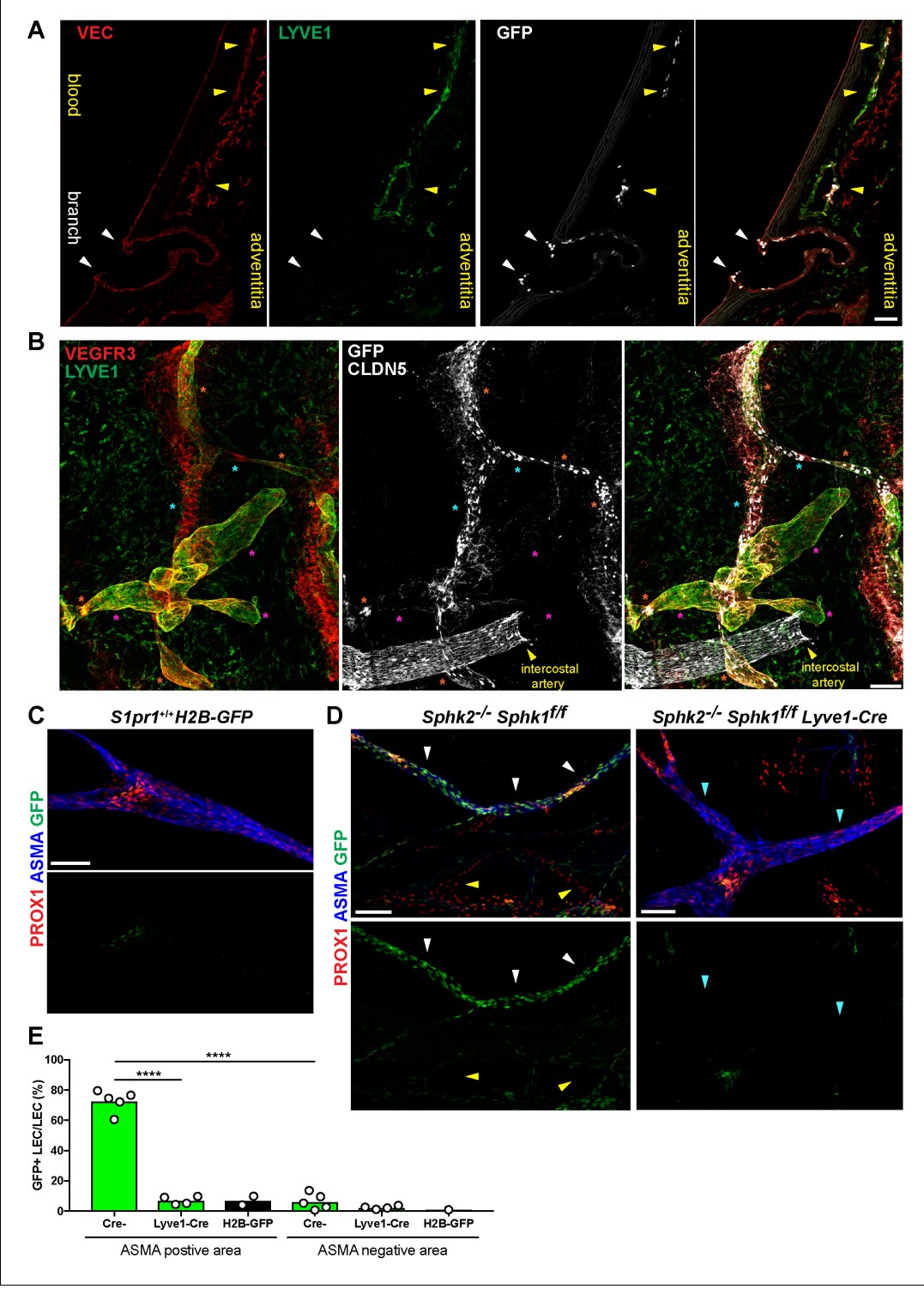

**Figure 8.** S1PR1/ß-arrestin coupling in aorta-associated lymphatic vessels and S1P-dependent signaling in LECs. (**A**) Representative confocal image of a sagittal cryosection (14 µM) of a S1RP1-GS mouse aorta immunostained for VEC and lymphatic vessel endothelial hyaluronic acid receptor 1 (LYVE1) (N = 2). White arrows indicate GFP+ arterial ECs at a branch point orifices and yellow arrows indicate adventitia-associated GFP+ LECs. Scale bar is 50 µM. (**B**) Confocal image of a S1RP1-GS mouse thoracic aorta whole-mount preparation immunostained for vascular endothelial growth factor receptor 3 (VEGFR3; *Flt4*), LYVE1, and CLDN5 with the tunica intima in contact with coverslip. Orange stars indicate VEGFR3+LYVE1+GFP+ areas, cyan stars indicate VEGFR3+LYVE1$^{low}$GFP+ areas, *Figure 8 continued on next page*

*Figure 8 continued*

and magenta stars indicate VEGFR3+LYVE1+GFP- areas. Image is representative of observations from three mice (see also *Figure 8—figure supplement 1B*). Scale bar is 100 μM. (C–D) Representative confocal images of mesenteric lymphatics from *H2B-GFP* control (C) and S1PR1-GS mice bearing *Sphk2⁻/⁻:Sphk1^(f/f)* (Cre-) or *Sphk2⁻/⁻: Sphk1^(f/f):Lyve1-Cre* (Lyve1-Cre+) alleles whole-mounted and immunostained for PROX1 and ASMA. White arrows indicate ASMA+PROX1+GFP+, yellow arrows indicate PROX1+GFP-, cyan arrows indicate ASMA+PROX1+GFP-. Scale bars are 100 μM. (E) Quantification of GFP+ signal over ASMA+PROX1+ and ASMA-PROX1+ areas from Cre- (n = 5), Lyve1-Cre (n = 4), and *H2B-GFP* control (n = 2) mice. One-way ANOVA followed by Bonferroni's *post hoc* test, ****$P < 0.0001$.

The online version of this article includes the following figure supplement(s) for figure 8:

**Figure supplement 1.** Expression of LEC transcripts and localization of ITGA6+ LEC and branch point arterial EC in close proximity.

---

cluster (Z-score >3 versus the remainder of ECs), while only 7% were enriched in vEC and/or aEC1-6 (*Figure 9A* and Suplementary File 6). None of the *S1pr1* ECKO down-regulated transcripts were enriched in the LEC cluster (see *Supplementary file 6*). A heatmap of the 78 LEC transcripts up-regulated in *S1pr1* ECKO MAECs is shown in *Figure 9B*. Among these were chemokine/cytokine pathway genes (*Irf8, Lbp, Il7, Il33 Ccl21, Tnfaip8l1*) as well lymphangiogenesis-associated genes (*Kdr, Prox1, Lyve1, Nr2f2*) (*Figure 9B* and *Figure 9—figure supplement 1A*). This suggests that loss of S1PR1 signaling in LECs alters transcriptional programs associated with lymphangiogenesis and inflammation/immunity.

Lymphatic vessels associated with the aorta and large arteries, although not well studied, are thought to be involved in key physiological and pathological processes in vascular and immune systems (*Csányi and Singla, 2019*; *Galkina and Ley, 2009*). For example, LEC expression of CCL21 mediates dendritic cell recruitment to lymphatic vessels during homeostasis and pathological conditions (*Vaahtomeri et al., 2017*). This chemokine was abundantly expressed in the LEC cluster (*Figure 8—figure supplement 1A*) and was up-regulated by S1PR1 loss-of-function (*Figure 9B* and *Figure 9—figure supplement 1A*). Whole-mount staining of S1PR1-GS mouse thoracic aortae for LYVE1 and CCL21 revealed aorta-associated CCL21+ lymphatics with high levels of S1PR1/ß-arrestin coupling (*Figure 9C and D*). We noted that CCL21 protein appeared as peri-nuclear puncta that likely marks the *trans*-Golgi network, as observed in dermal LECs (*Vaahtomeri et al., 2017*). Sagittal sections of the thoracic aorta indicate that GFP^high, LYVE1⁺ adventitial lymphatics express CCL21 protein (*Figure 9D*), suggesting that ß-arrestin-dependent down-regulation of S1PR1 correlates with CCL21 expression (*Figure 9D*).

These studies show that S1PR1/ß-arrestin coupling in a subset of adventitial lymphatics correlates with S1PR1-mediated attenuation of lymphagiogenic/inflammatory gene expression. We observed that the fraction of PDPN⁺ LECs was not altered between *S1pr1* ECKO and WT aorta tissues (*Figure 9—figure supplement 1B*), indicating that there is not widespread lymphagiogenesis or LEC proliferation. However, analysis of *Flt4* vs. *Pdpn* expression and *Ccl21a* vs. *Pdpn* expression revealed the presence of LECs expressing *Flt4* and *Ccl21a* but not *Pdpn* (*Figure 9—figure supplement 2*). Therefore, it is possible that *S1pr1* ECKO leads to expansion of a PDPN^low population in the adventitia. Nonetheless, these data indicate that S1PR1 mediates gene expression in adventitial lymphatics of the adult mouse aorta under homeostasis.

## Discussion

Intracellular signaling through G protein- and ß-arrestin-dependent pathways is tightly regulated at the levels of GPCR expression and ligand availability. Endothelium of major organs, such as brain, lung, skeletal muscle, and the aorta, express unique sets of GPCRs with only five receptors commonly expressed, one of which is *S1pr1* (*Kaur et al., 2017*). Despite ubiquitoius endothelial *S1pr1* expression, S1PR1/ß-arrestin coupling in vivo, as reported by GFP in S1PR1-GS mice, revealed heterogeneous signaling in multiple organs (*Kono et al., 2014*; *Galvani et al., 2015*). Here, we describe high frequency of aortic GFP+ (i.e. S1PR1/ß-arrestin coupling) ECs around intercostal branch point orifices and heterogeneous GFP+ ECs throughout the remainder of intimal aortic endothelium. We and others have shown that endothelial ablation of *S1pr1* (*S1pr1* ECKO) or reduction of circulatory

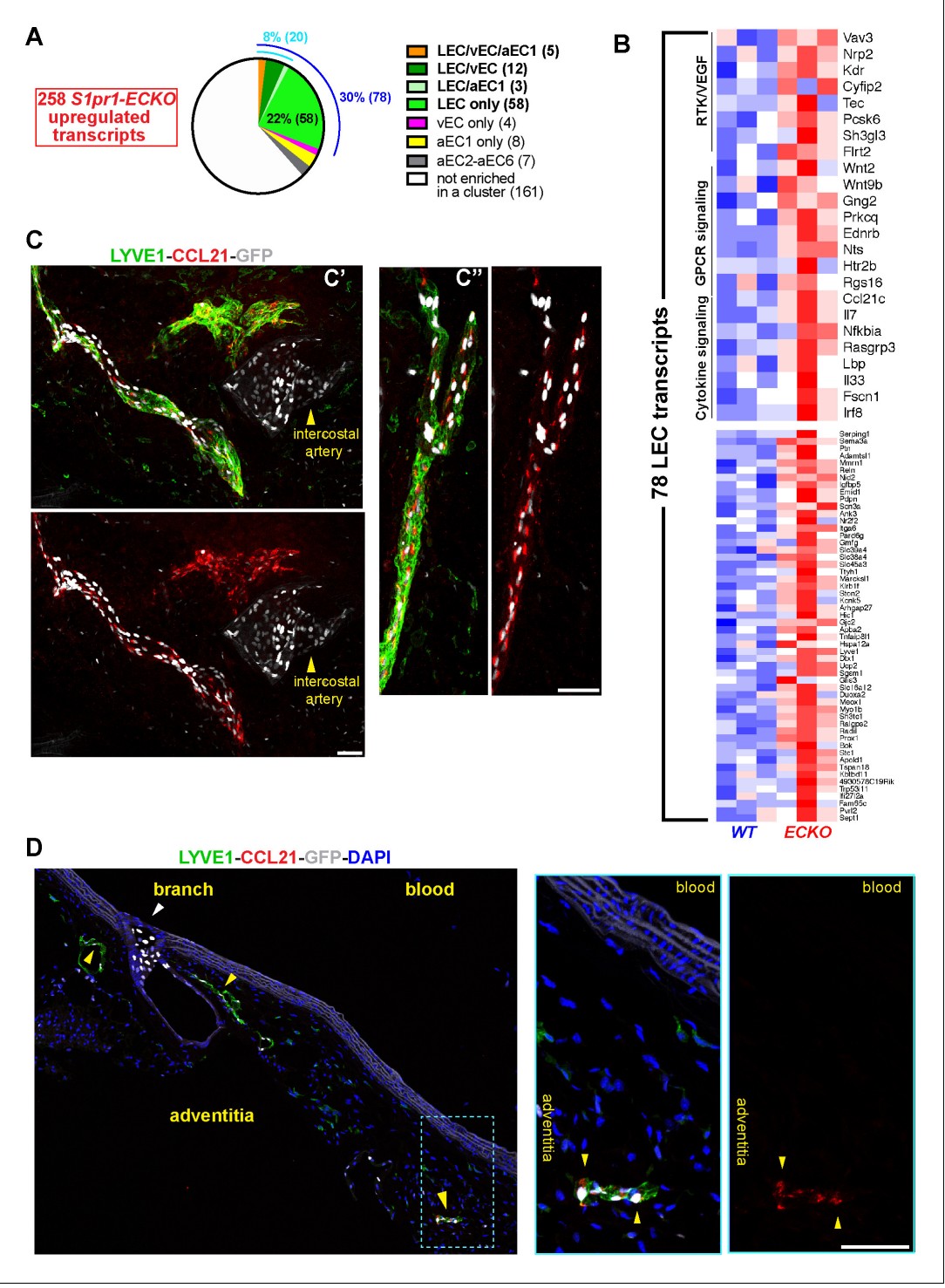

**Figure 9.** S1PR1 regulation of gene expression in aorta-associated LECs. (A) Pie-chart distribution of *S1pr1* ECKO MAECs up-regulated transcripts. The 258 transcripts were binned according to their cluster assignment from scRNA-seq analysis. The blue and cyan numbers indicate the total percentage of transcripts enriched in LEC and LEC plus at least one cluster, respectively. The numbers in parentheses are absolute transcripts numbers. (See *Supplementary file 6*) (B) Heatmap (row Z-scores) of all 78 *S1pr1* ECKO MAECs up-regulated transcripts that are LEC-enriched. Selected GSEA pathways are identified in the top heatmap. (See *Supplementary file 5*) (C) Confocal images of two fields (c' and c'') of a whole-mounted S1RP1-GS mouse thoracic aorta immunostained for LYVE1 and C-C motif chemokine 21 (CCL21). The tunica adventitia was facing the coverslip. (D) Confocal images of a sagittal cryosection (14 µM) of an S1RP1-GS mouse thoracic aorta immunostained CCL21 and LYVE1. Yellow

*Figure 9 continued on next page*

*Figure 9 continued*

arrows indicate adventitia-associated CCL21+LYVE1+GFP+ cells and the white arrow indicate GFP+LYVE1- cells at a branch. Scale bars are 50 μM.

The online version of this article includes the following figure supplement(s) for figure 9:

**Figure supplement 1.** Up-regulation of LEC transcripts in *S1pr1* ECKO MAECs is not associated with a change in the proportion of aorta-associated PDPN$^+$ LECs.

**Figure supplement 2.** Heterogeneous LEC marker gene expression in aorta-associated LECs.

S1P disrupts endothelial barriers (*Camerer et al., 2009*; *Christensen et al., 2016*; *Christoffersen et al., 2011*; *Oo et al., 2011*; *Yanagida et al., 2017*). Furthermore, the descending aorta of *S1pr1* ECKO mice displayed exacerbated plaque formation in the *Apoe*$^{-/-}$Western diet (WD)-induced atherosclerosis model (*Galvani et al., 2015*). However, we lack information regarding EC transcriptional responses that correlate with or are directly downstream of S1PR1 signaling. Here, we profiled the transcriptomes and open chromatin landscapes of aortic ECs with high (GFP$^{high}$) or low (GFP$^{low}$) levels of S1PR1/ß-arrestin coupling, as well as *S1pr1* ECKO aortic ECs.

*S1pr1* ECKO MAECs up-regulated transcripts in TNFα/cytokine signaling pathways and exhibited enhanced chromatin accessibility at NFκB binding sites. Concomitantly, the glucocorticoid receptor pathway was suppressed. These mRNA and chromatin signatures were shared between *S1pr1* ECKO and GFP$^{high}$ MAECs, suggesting that persistent ß-arrestin recruitment to S1PR1 can result in down-regulation of membrane-localized S1PR1 and a subsequent loss-of-function phenotype.

There were many (2,145) DEGs between GFP$^{high}$ and GFP$^{low}$ MAECs, but relatively few (365) between ECKO and WT MAECs, which suggested that the GFP$^{high}$ and/or GFP$^{low}$ populations reported aortic EC subtypes in addition to S1PR1-regulated transcripts. Indeed, chromatin regions uniquely open in GFP$^{high}$ MAECs were enriched with binding sites for TFs that have well-defined but divergent roles in endothelial cells, such as SOX17 (arterial) (*Corada et al., 2013*; *Zhou et al., 2015*) and COUP-TFII (venous/lymphatic) (*Lindskog et al., 2014*). Nonetheless, we present the first collection of putative regulatory regions from freshly isolated mouse aortic ECs, which is a critical dataset for future studies of individual enhancer functionalities.

Our scRNA-seq analysis addressed with high resolution the heterogeneity among MAECs. We identified six arterial EC clusters (aEC1-6), one lymphatic EC cluster (LEC) and one venous EC cluster (vEC). Immunohistochemical analyses revealed LEC cells as including lymphatic structures of the aortic adventita, aEC1 cells as circumscribing intercostal branch point orifices, and aEC2 cells as heterogeneously dispersed throughout intimal endothelium. Each of these clusters exhibited a high frequency (>90%) of GFP$^{high}$ EC. We also describe aEC3 cells, which contained comparatively few (<60%) GFP$^{high}$ ECs. aEC3 cells strongly resembled an *Atf3*-positive cluster reported by *McDonald et al. (2018)* that mediates endothelial regeneration (*McDonald et al., 2018*). Considering that *Atf3*-positive ECs were absent in old (18 month) mice (*McDonald et al., 2018*), we hypothesize that aEC3-like cells disappear over time while aEC2-like cells increase in frequency in the aorta intima. This notion is supported by the higher frequency of non-branch point GFP$^{high}$and TSP1-expressing intimal ECs in adult relative to P6 and P12 mice.

When compared with findings from two recent aorta scRNA-seq studies (*Kalluri et al., 2019*; *Lukowski et al., 2019*), our clustering segregated vEC, LEC, and aEC1 from aEC2-6. We suspect that use of S1PR1-GS mice facilitated deconvolution of LEC and vEC from the distinct aEC1 population. Despite their proximity to intercostal branch points, aEC1 cells do not exhibit a transcriptomic signature prototypical of inflammation, as might be expected of ECs in an environment with disturbed flow (*Chiu and Chien, 2011*). Furthermore, high levels of S1PR1/ß-arrestin coupling and expression of unique genes (e.g. *Itga6*) in aEC1 cells are independent of both circulatory S1P and age in postnatal mice. To further explore temporal regulation of cluster-specific genes, we examined scRNA-seq of FACS-sorted VEGFR2$^{high}$ cells from E8.25 embryos (*Pijuan-Sala et al., 2019*) and the endothelial cluster from scRNA-seq of E9.5 to E13.5 embryos (*Cao et al., 2019*). These embryonic cells exhibited expression of aEC1-enriched transcripts (*Lef1, Itga6, Alpl, Flt1, Igf2*), but depletion of aEC2-6-enriched transcripts (*Thbs1, Sod3, Vcam1, Sfrp1, Pcolce2, Dcn*). Together with embryonic EC gene expression data (*Figure 7—figure supplement 3*) and immunohistochemical analysis of E11.5 S1PR1-GS embryos, our data suggest that aEC1 cells are more characteristic of embryonic EC

than are the majority of intimal ECs. Similarly, vEC/LEC-specific transcripts (*Prox1*, *Lyve1*, *Nr2f2*) were also expressed in embryonic ECs in both studies (*Pijuan-Sala et al., 2019*; *Cao et al., 2019*). Thus, transcriptomic similarities between aEC1 and LEC/vEC in the adult aorta may be retained from development, perhaps through epigenetic modifications common between these cell types. It is also possible that the unique anatomical location of aEC1 (at the circumferential ridge at aortic branch points) may promote a distinct EC phenotype either because of spatial positioning or environmental factors.

*S1pr1* ECKO and *S1pr1*$^{-/-}$ animals display an aortic hyper-branching phenotype between E11.5 and E13.5 that is incompatible with life after E14.5 (*Gaengel et al., 2012*). Therefore, S1PR1 is required for normal embryonic branching morphogenesis. Consistently, E9.5-E10.5 S1PR1-GS embryos show high GFP expression (S1PR1/ß-arrestin coupling) in the dorsal aorta (*Kono et al., 2014*). This contrasts with adult aortae, wherein the highest levels of S1PR1/ß-arrestin coupling are concentrated around the orifices of intercostal branch points. These findings further suggest that the unique gene expression program of aEC1 cells is established during morphogenesis of intercostal arteries during development. It is possible that the chromatin accessibility differences that we observed near aEC1-enriched and –depleted genes were established during embryonic or early postnatal life and were retained into adulthood. While we cannot exclude shear forces as contributing to the aEC1 transcriptional program, our data suggest that disturbed flow is not the main driver of gene expression in these cells.

The extent to which aEC1 cells at branch point orifices are functionally distinct remains to be determined. Identification of ITGA6 as a marker of this population will facilitate future studies. For example, combinations of pan-EC, LEC, and ITGA6 antibodies can be used to purify or enrich this population in developmental or disease models (e.g. atherosclerosis). Moreover, *cis*-elements proximal to aEC1-specific genes can be applied to the 'Dre-rox/Cre-loxP' system (*Pu et al., 2018*) to specifically manipulate gene expression in aEC1 cells.

Non-branch point (i.e. aEC2) cells require circulatory S1P for S1PR1/ß-arrestin coupling. Similarly, mesenteric LECs required S1P in lymph for normal levels of S1PR1/ß-arrestin coupling. Notably, we did not detect *S1pr1* ECKO down-regulated transcripts in LEC cluster, but we did detect five such transcripts in aEC1 cells (*Dusp26*, *Enah*, *Eps8l2*, *Hapln1* and *S1pr1*) and two transcripts in aEC2 cells (*Nfix*, *Znrf2*). Among the clusters identified in this study, LEC-enriched transcripts were affected the most upon EC ablation of *S1pr1* (Suplementary File 6). This suggests that loss of S1P/S1PR1 signaling either alters cell-intrisic phenotypes of peri-aortic LECs or induces expansion of one or multiple LEC subtypes.

There is accumulating direct and indirect evidence for key roles of adventital lymphatics in atherogenesis (*Csányi and Singla, 2019*; *Maiellaro and Taylor, 2007*). For example, auto-antibodies against oxidized LDL (OxLDL) inhibit macrophage OxLDL uptake and mitigate atherosclerosis (*Shaw et al., 2000*). This implies that antigen presenting cells (APCs) phagocytose OxLDL epitopes, then travel via adventital lymphatics to lymphoid organs (e.g. lymph nodes) and present OxLDL antigens to B cells. Murine atherosclerotic lesions were found to harbor 'atypical, lymphatic-like' capillaries that were VEGFR3+ but LYVE1- (*Taher et al., 2016*), which is consistent with our observations of adventitial LEC heterogeneity. Considering the critical role of lymphatic EC-derived CCL21 in regulating the trafficking of APCs (*Vaahtomeri et al., 2017*), and perhaps other adaptive immune cells, there is an impetus to determine the extent to which adventitial lymphatics are a viable target for atherosclerosis therapy.

Identification of embryonic LEC with high S1PR1/ß-arrestin coupling may suggest a functional role for S1PR1 in developmental lymphangiogenesis. We speculate that future studies of *S1pr1*$^{f/f/}$ mice bearing LEC-specific Cre-drivers will provide insight into S1PR1-mediated events during developmental lymphangiogenesis.

While there is scant information about the roles of S1P/S1PR1 signaling in adult lymphatic vasculature, our findings lay a groundwork for future studies of S1PR1-mediated LEC phenotype regulation in homeostatic processes and inflammatory/autoimmune diseases. A recent study found that lymph-derived S1P facilitates CCL21 deposition in high endothelial venules and dendritic cell recruitment (*Simmons et al., 2019*). While *S1pr1* ECKO animals exhibit exacerbated atherosclerosis (*Galvani et al., 2015*), we cannot discern whether this was due to phenotypes of lymphatic ECs, arterial ECs, or both cell types. Future studies should use artery- and lymphatic-specific Cre-drivers to distinguish between the roles of S1PR1 in different types of vasculature. Such mechanistic studies

will help to determine the utility of S1PR1 modulators in treating lymphatic-mediated vasculopathies.

# Materials and methods

**Key resources table**

| Reagent type (species) or resource | Designation | Source or reference | Identifiers | Additional information |
|---|---|---|---|---|
| Genetic reagent (*M. musculus*) | S1pr1$^{flox/flox}$: S1pr1$^{tm2Rlp}$ | *Allende et al. (2003)* | RRID:MGI:2681963 | *S1pr1* ECKO |
| Genetic reagent (*M. musculus*) | B6N.129S6(FVB)-S1pr1$^{tm3.1(tTA,-Arrb2)Rlp}$/J | The Jackson Laboratory, *Kono et al. (2014)* | RRID:IMSR_JAX:026275 | S1PR1-GS |
| Genetic reagent (*M. musculus*) | Cdh5-Cre$^{ERT2}$: Tg(Cdh5-cre/ERT2)1Rha | *Sörensen et al., 2009* | RRID:MGI:3848984 | *S1pr1* ECKO |
| Genetic reagent (*M. musculus*) | Tg(tetO-HIST1H2BJ/GFP)47Efu/J | The Jackson Laboratory | RRID:IMSR_JAX:005104 | S1PR1-GS |
| Genetic reagent (*M. musculus*) | B6;129P2-Lyve1$^{tm1.1(EGFP/cre)}$Cys/J | The Jackson Laboratory, *Pham et al., 2010* | RRID:IMSR_JAX:012601 | S1PR1-GS-Lymph-S1P-less |
| Genetic reagent (*M. musculus*) | Sphk1$^{tm2Cgh}$ | *Pham et al., 2010* | MGI:3707997 | S1PR1-GS-Lymph-S1P-less |
| Genetic reagent (*M. musculus*) | Sphk2$^{tm1.1Cgh}$ | *Pham et al., 2010* | MGI:3708000 | S1PR1-GS-Lymph-S1P-less |
| Genetic reagent (*M. musculus*) | Rosa26-Cre$^{ERT2}$: Gt(ROSA)26Sortm1 (cre/ERT2)Alj | *Takeda et al., 2007* | MGI:3778915 | S1PR1-GS-S1P-less |
| Antibody | (PE)-conjugated anti-CD31, (MEC13.3, Rat Monoclonal) | Biolegend | Cat #: 102508 RRID:AB_312915 | |
| Antibody | (APC)-conjugated anti-mouse CD45 (30-F11, Rat Monoclonal) | Biolegend | Cat #: 103112 RRID:AB_312977 | |
| Antibody | (APC)-conjugated anti-mouse TER119/ Erythroid Cells (Rat Monoclonal) | Biolegend | Cat #:116212 RRID:AB_313713 | |
| Antibody | Anti-Mouse CD16/CD32 (Rat Monoclonal) | Thermo Fisher Scientific, eBioscience | Cat #: 14-0161-82 RRID:AB_467133 | |
| Antibody | Anti-mouse VE-cadherin (goat polyclonal) | R and D Systems | Cat #: AF1002 RRID:AB_2077789 | |
| Antibody | Anti-mouse Fibrinogen (goat polyclonal) | Accurate Chemical | Cat #: YNGMFBG | |
| Antibody | Anti-Ki-67 (SolA15, rat monoclonal) | Thermo Fisher Scientific, eBioscience | Cat #: 14-5698-80 RRID:AB_10853185 | |
| Antibody | Anti-mouse LYVE1 (rabbit Polyclonal) | ReliaTech | Cat #: 103-PA50AG RRID:AB_2783787 | 1:300 |
| Antibody | Anti-mouse LYVE1 (goat polyclonal) | R and D systems | Cat #: AF2125 RRID:AB_2297188 | 1:300 |
| Antibody | Anti-mouse VEGFR3/Flt-4 (goat polyclonal) | R and D systems | Cat #: AF743 RRID:AB_355563 | 1:200 |

*Continued on next page*

*Continued*

| Reagent type (species) or resource | Designation | Source or reference | Identifiers | Additional information |
|---|---|---|---|---|
| Antibody | Anti-human/mouse CD49f (ITGA6) (GoH3, rat monoclonal) | BioLegend | Cat #: 313602 RRID:AB_345296 | 1:200 |
| Antibody | Anti-mouse ALPL (goat polyclonal) | R and D systems | Cat #: AF2910 RRID:AB_664062 | 1:200 |
| Antibody | Biotinylated anti-TSP1 (A6.1, mouse monoclonal) | Thermo Fisher Scientific | Cat #: MA5-13395 RRID:AB_10982819 | 1:200 |
| Antibody | Alexa Fluor 488-conjugated anti-CLDN5 (4C3C2, mouse monoclonal) | Thermo Fisher Scientific, Invitrogen | Cat #: 352588 RRID:AB_2532189 | 1:100 |
| Antibody | Anti-FSP1/S100A4 (rabbit polyclonal) | Millipore Sigma | Cat #: 07–2274 RRID:AB_10807552 | 1:300 |
| Antibody | anti-mouse/rat CD31/PECAM-1 (goat polyclonal) | R and D systems | Cat #: AF3628 RRID:AB_2161028 | 1:300 |
| Antibody | anti-CD31 (SZ31, rat monoclonal) | HistoBiotec | Cat #: DIA-310 RRID:AB_2631039 | 1:100 |
| Antibody | anti-LEF1 (C12A5, rabbit monoclonal) | Cell Signaling Technologies | Cat #: 2230 RRID:AB_823558 | 1:200 |
| Antibody | anti-mouse VCAM1 (M/K, rat monoclonal) | Millipore Sigma | Cat #: CBL1300 RRID:AB_2214062 | 1:200 |
| Antibody | anti-mouse NOGGIN (goat polyclonal) | R and D systems | Cat #: AF719 RRID:AB_2151669 | 1:200 |
| Antibody | anti-mouse CCL21/6Ckine (goat polyclonal) | R and D systems | Cat #: AF457 RRID:AB_2072083 | 1:200 |
| Antibody | anti-human PROX1 (goat polyclonal) | R and D systems | Cat #: AF2727 RRID:AB_2170716 | 1:200 |
| Antibody | anti-jellyfish GFP (chicken polyclonal) | ThermoFisher | Cat #: A10262 RRID:AB_2534023 | 1:500 |
| Antibody | Cy3-conjugated anti-human alpha smooth muscle actin (1A4, mouse monoclonal) | Sigma-Aldrich | Cat #: C6198 RRID:AB_476856 | 1:300 |
| Commercial assay or kit | RNeasy Micro Kit | Qiagen | Cat #: 74004 | |
| Commercial assay or kit | SMART-Seq2 v4 Ultra Low RNA Kit for Sequencing | TakaraBio | Cat #: 634888 | |
| Commercial assay or kit | High Sensitivity RNA ScreenTape | Agilent | Cat #: 5067–5579 | |
| Commercial assay or kit | High Sensitivity D1000 ScreenTape | Agilent | Cat #: 5067–5584 | |
| Commercial assay or kit | Nextera XT2 DNA Library Prep Kit for RNA-seq | Illumina | Cat #: FC-131–1024 | |
| Commercial assay or kit | Nextera DNA Library Prep Kit for ATAC-seq (buffer TD and TDE1 enzyme) | Illumina | Cat #: FC-121–1030 | |
| Commercial assay or kit | MinElute Reaction Cleanup Kit | Qiagen | Cat #: 28204 | |

*Continued on next page*

Continued

| Reagent type (species) or resource | Designation | Source or reference | Identifiers | Additional information |
|---|---|---|---|---|
| Commercial assay or kit | MinElute PCR Purification Kit | Qiagen | Cat #: 28004 | |
| Commercial assay or kit | NEBNext High-Fidelity 2X PCR Master Mix | New England Biolabs | Cat#: M0541S | |
| Chemical compound or drug | Tamoxifen | Sigma-Aldrich | Cat #: T5648 | |
| Chemical compound or drug | Corn oil | Sigma-Aldrich | Cat #: C8267 | |
| Chemical compound or drug | Fluoromount-G slide mounting medium | Southern Biotech | Cat #: 0100–01 | |
| Chemical compound or drug | ProLong Gold | Thermo Fisher Scientific, Invitrogen | P36934 | |
| Chemical compound or drug | Liberase TM | Sigma-Aldrich | Cat #: 5401127001 | |
| Chemical compound or drug | deoxyribonuclease I from bovine pancreas, type 2 | Sigma-Aldrich | Cat #: D4527 | |
| Chemical compound or drug | Bovine Serum Albumin lyophilized powder, essentially fatty acid free,≥96% (agarose gel electrophoresis) | Sigma-Aldrich | Cat #: A6003 | |
| Chemical compound or drug | 2-MercaptoEthanol | Sigma-Aldrich | Cat #: M3148 | |
| Chemical compound or drug | Digitonin (20 mg/ml) | Promega | Cat #: G9441 | |
| Chemical compound or drug | Donkey Serum | Sigma-Aldrich | Cat #: D9663 | |
| Chemical compound or drug | RNase inhibitor | Takara Bio | Cat #: 2313B | |
| Chemical compound or drug | Sphingosine-1-Phosphate (d18:1) | Avanti lipids | Cat #: 860492P | |
| Sequence-based reagent | Smart-dT30VN | Sigma-Aldrich | | |
| Sequence-based reagent | ERCC RNA Spike-In Mix | Thermo Fisher Scientific, Ambion | Cat #: 4456740 | |
| Software, algorithm | Fiji | NIH | https://imagej.net/Fiji RRID:SCR_002285 | |
| Software, algorithm | Graphpad Prism 8.0 | Graphpad Software | https://www.graphpad.com/scientific-software/prism/ RRID:SCR_002798 | |
| Software, algorithm | STAR | PMID: 23104886 | https://github.com/alexdobin/STAR RRID:SCR_015899 | |
| Software, algorithm | Rsubread | PMID: 23558742 | https://bioconductor.org/packages/release/bioc/html/Rsubread.html RRID:SCR_016945 | |
| Software, algorithm | edgeR | PMID: 19910308 | https://bioconductor.org/packages/release/bioc/html/edgeR.html RRID:SCR_012802 | |

*Continued*

| Reagent type (species) or resource | Designation | Source or reference | Identifiers | Additional information |
|---|---|---|---|---|
| Software, algorithm | RSEM | PMID: 21816040 | https://deweylab.github.io/RSEM/ RRID:SCR_013027 | |
| Software, algorithm | bowtie2 | PMID: 22388286 | http://bowtie-bio.sourceforge.net/bowtie2/index.shtml RRID:SCR_005476 | |
| Software, algorithm | Picard | Broad Institute | https://broadinstitute.github.io/picard/ RRID:SCR_006525 | |
| Software, algorithm | MACS2 | PMID: 18798982 | https://github.com/taoliu/MACS RRID:SCR_013291 | |
| Software, algorithm | Bedtools | PMID: 20110278 | https://bedtools.readthedocs.io/en/latest/ RRID:SCR_006646 | |
| Software, algorithm | bedops | PMID: 22576172 | https://bedops.readthedocs.io/en/latest/ RRID:SCR_012865 | |
| Software, algorithm | Hypergeometric Optimization of Motif EnRichment (HOMER) | PMID: 20513432 | http://homer.ucsd.edu/homer/ RRID:SCR_010881 | |
| Software, algorithm | DeepTools | PMID: 27079975 | https://deeptools.readthedocs.io/en/develop/ RRID:SCR_016366 | |
| Software, algorithm | Integrative Genomics Viewer | PMID: 21221095 | http://software.broadinstitute.org/software/igv/ RRID:SCR_011793 | |
| Software, algorithm | Samtools | PMID: 19505943 | http://www.htslib.org/ RRID:SCR_002105 | |
| Software, algorithm | velocyto | PMID: 30089906 | http://velocyto.org/velocyto.py/ | |
| Software, algorithm | Pagoda2 | PMID: 26780092 PMID: 30089906 | https://github.com/hms-dbmi/pagoda2 http://pklab.med.harvard.edu/nikolas/pagoda2/frontend/current/pagodaLocal/ RRID:SCR_017094 | Binary (.bin) file in *Supplementary file 7* was generated as instructed here: https://github.com/hms-dbmi/pagoda2/blob/master/vignettes/pagoda2.walkthrough.oct2018.md |

## Mice

Animal experiment protocols were approved by the Institutional Animal Care and Use Committees (IACUC) of Boston Children's Hospital and the French Department of Education. S1PR1-GS mice were previously reported (*Kono et al., 2014*). S1PR1-GS mice used for experiments harbored a single *S1pr1*$^{knockin}$ (*S1pr1-tTA-IRES-mArrb2-TEV*) allele (*S1pr1*$^{ki/+}$) as well as a single *H2B-GFP* reporter allele. *S1pr1*$^{f/f}$ mice (*Allende et al., 2003*) were bred with *Cdh5-Cre*$^{ERT2}$ mice (*Sörensen et al., 2009*) to generate *S1pr1* ECKO mice, as previously described (*Galvani et al., 2015*; *Jung et al., 2012*). Gene deletion was achieved by intraperitoneal injection of tamoxifen (2 mg/day) at 5–6 weeks of age for five consecutive days. Tamoxifen treated mice were rested for a minimum of 2 weeks prior to experiments.

S1PR1-GS mice deficient in LEC S1P production were generated by excising a conditional knockout allele for *Sphk1* in an *Sphk2* knockout background with *Lyve1-Cre* (*S1pr1*$^{ki/+}$:*Sphk1*$^{f/f}$:*Sphk2*$^{-/-}$:*Lyve1-Cre*$^{+}$) essentially as described by Cyster and colleagues (*Pham et al., 2010*) and excision efficiency confirmed by the induction of lymphopenia. S1PR1-GS mice deficient in plasma S1P (S1PR1-GS-S1P-less) were generated by crossing S1PR1-GS mice with *Sphk1*$^{f/f}$:*Sphk2*$^{-/-}$:*Rosa26-Cre-ER*$^{T2}$ mice to obtain *S1pr1*$^{ki/+}$:*GFP*$^{+}$:*Sphk1*$^{f/f}$:*Sphk2*$^{-/-}$:*Rosa26-Cre-ER*$^{T2+}$ mice (the *Rosa26-Cre-ER*$^{T2}$ allele

is described in *Takeda et al. (2007)*. Tamoxifen was administered to S1PR1-GS-S1P-less mice and Cre- littermate controls as described above. Experiments were performed between 23 and 25 weeks after the final tamoxifen dose.

Young adult (aged 8 to 12 weeks) males and females were used for sequencing experiments. Males and females of similar age (7 to 18 weeks) were used for imaging studies, unless indicated otherwise. For examination of VCAM1 in *Figure 5—figure supplement 2B*, 200 µL lipopolysaccharide (Sigma-Aldrich, L2630) in PBS was injected i.p. (5.5 mg/kg) followed by euthanasia and tissue harvest after 9 hr. For timed matings, embryonic day (E) 0.5 was defined as noon on the date of the vaginal plug and embryos were harvested at E11.5.

## Lymphocyte counts

Blood was drawn from the retroorbital venous plexus into EDTA tubes and blood cells enumerated with a Hemavet 950 cell counter (Drew Scientific).

## Lung vascular leakage assay

Lung vascular integrity was assessed by administration of 6 µL/g 0.5% Evans Blue dye (Sigma # E2129) via the retro-orbital venous plexus. Two hours later, lungs were perfused via the right ventricle with 10 mL heparin-DPBS, harvested, and Evans Blue was extracted overnight at 55°C in formamide. Lung accumulation of Evans Blue was quantified by measuring absorbance of the lung extract at 620 nm and expressed as corrected for absorbance at 740 nm.

## FACS isolation and single cell sequencing of mouse aortic endothelial cells

After $CO_2$ euthanasia, the right atrium was opened and the left ventricle was perfused with 10 mL phosphate-buffered saline (PBS) (Corning). Aortae were dissected from the root to below the common iliac bifurcation and transferred into ice ice-cold 1x HBSS (Sigma-Aldrich, H1641). Whole aortae were incubated in HBSS containing elastase (4.6 U/mL, LS002292, Worthington), dispase II (1.3 U/mL, Roche), and hyaluronidase (50.5 U/mL, Sigma-Aldrich, H3506) at 37 °C for 10 min in wells of a 6-well plate. Aortae were then transferred to a 100 mm dish with 1 mL HBSS and minced using small scissors. Minced aortae were transferred to a low protein binding 5 mL tube containing Liberase (0.6 U/mL, Sigma-Aldrich), collagenase II (86.7 U/mL, LS004174, Worthington), and DNase (62.0 U/mL, Sigma-Aldrich, D4527) in 4.3 mL HBSS and incubated at 37 ° C for 40 min with rotation in a hybridization oven. The cell suspension was then triturated 10 times through an 18 G needle to dissociate clumps, followed by addition of 400 µL STOP solution (3 mM EDTA, 0.5% fatty acid-free bovin serum albumin (FAF-BSA) (Sigma-Aldrich, A6003) in 1x HBSS). For the remainder of the procedure, cells were kept on ice and all centrifugation steps were performed at 4 °C.

Cells were spun at 500 xg for 5 min, the supernatant was removed, then cells were washed with 4 mL STOP solution and spun at 500 xg for 5 min. The supernatant was removed, then cells were washed with 4 mL blocking solution (0.25% FAF-BSA in HBSS) and filtered through FACS tubes with filter caps (Falcon). After centrifugation and supernatant removal, cells were stained with phycoerythrin (PE)-conjugated anti-mouse CD31 (MEC13.3, Biolegend, San Diego, CA), allophycocyanin (APC)-conjugated anti-mouse CD45 (30-F11, Biolegend) and APC-conjugated TER119 (Biolegend, 116212) antibodies in blocking solution with anti-CD16/32 (2.5 µg/mL) for 45 min on ice. DAPI (0.7 µM) was added for the final 5 min of staining to exclude dead cells. Aortic cells were washed with 4.5 mL FACS buffer (0.25% FAF-BSA in PBS) before sorting for selection of CD31$^+$/CD45$^-$/TER119$^-$/GFP$^{high}$ and CD31$^+$/CD45$^-$/TER119$^-$/GFP$^{low}$ cells using BD FACSAria II (BD Bioscience) (see *Figure 1B*). Cells from *S1pr1* WT and *-ECKO* mice were sorted using the GFP$^{low}$ gate (see *Figure 1—figure supplement 1A*) because it includes MAECs from mice genetically negative for the *H2B-GFP* reporter allele and stained with the same antibody panel. Cells from 2 to 4 aortae of age and sex-matched adult mice were pooled for each individual experiment (ATAC-seq, RNA-seq and scRNA-seq). Cells were sorted into either 0.1% FAF-BSA/PBS or buffer RLT (Qiagen) supplemented with β-mercaptoethanol for ATAC-seq and RNA-seq, respectively.

For single-cell RNA-seq, GFP$^{high}$ and GFP$^{low}$ cells were gated as described above. Library preparation from single cells was performed as previously described (*Vanlandewijck et al., 2018*). Briefly, cells were deposited into individual wells of 384-well plates containing 2.3 µL of lysis buffer (0.2%

Triton-X (Sigma-Aldrich, T9284), 2 U/μL RNase inhibitor (ClonTech, 2313B), 2 mM dNTP's (Thermo-Fisher Scientific, R1122), 1 μM Smart-dT30VN (Sigma-Aldrich), ERCC 1:4 × 10⁷ dilution (Ambion, # 4456740)) prior to library preparation using the Smart-seq2 protocol (*Picelli et al., 2014*).

## Bulk RNA-seq and analysis

Cells sorted into buffer RLT were subjected to total RNA extraction using the RNeasy Micro Kit (Qiagen). The High Sensitivity RNA ScreenTape (Agilent) was used to verify RNA quality before synthesis of double-stranded cDNA from 5 to 10 ng RNA using the SMART-Seq2 v4 Ultra Low RNA Kit for Sequencing (Takara Bio) according to the manufacturer's instructions. Agilent 2100 Bioanalyzer and High Sensitivity DNA Kit (Agilent) were used to verify cDNA quality. cDNA libraries were prepared for sequencing using the Illumina Nextera XT2 kit (Illumina), and ~20–40 million paired-end reads (2 × 75 bp) were sequenced for each sample.

Reads from each sample were aligned to the MGSCv37 (mm9) genome assembly using STAR (*Dobin et al., 2013*) with the options: `–runMode alignReads –outFilterType BySJout –out-FilterMultimapNmax` 20 `–alignSJoverhangMin` 8 `–alignSJDBoverhangMin` 1 `–outFilter-MismatchNmax` 999 `–alignIntronMin` 10 `–alignIntronMax` 1000000 `–alignMatesGapMax` 1000000 `–outSAMtype BAM SortedByCoordinate –quantMode TranscriptomeSAM`. Gene-level counts over UCSC annotated exons were calculated using the Rsubread package and 'feature-Counts' script (*Liao et al., 2013*) with options: -M –O –p –d 30 –D 50000. The resultant count table was input to edgeR (*Robinson et al., 2010*) for differential gene expression analysis. The. bam files from STAR were input to the RSEM (*Li and Dewey, 2011*) script 'rsem-calculate-expression' with default parameters to generate FPKMs for each replicate.

## ATAC-seq and analysis

ATAC-seq libraries were prepared according to the previously described fast-ATAC protocol (*Corces et al., 2016*). Briefly, 800–4,000 FACS-isolated cells in 0.1% FAF-BSA/PBS were pelleted by centrifugation at 400 ×*g* at 4°C for 5 min. Supernatant was carefully removed to leave the cell pellet undisturbed, then cells were washed once with 1 mL ice-cold PBS. The transposition mix [25 μL buffer TD, 2.5 μL TDE1 (both from Illumina FC-121–1030), 1 μL of 0.5% digitonin (Promega, G9441) and 16 μl nuclease-free water] was prepared and mixed by pipetting, then added to the cell pellet. Pellets were disrupted by gently flicking the tubes, followed by incubation at 37°C for 30 min in an Eppendorf ThermoMixer with constant agitation at 300 rpm. Tagmented DNA was purified using the MinElute Reaction Cleanup Kit (Qiagen, 28204), and subjected to cycle-limiting PCR as previously described (*Buenrostro et al., 2013*). Transposed fragments were purified using the MinElute PCR Purification Kit (Qiagen, 28004) and Agilent DNA Tapestation D1000 High Sensitivity chips (Agilent) were used to quantify libraries. ~ 20–60 million paired-end reads (2 × 75 bp) were sequenced for each sample on a NextSeq instrument (Illumina).

Read alignment to the MGSCv37 (mm9) genome assembly was performed with bowtie2 (*Langmead and Salzberg, 2012*) and the options: `–very-sensitive` –X 2000 `–no-mixed –no-dis-cordant`. Duplicated fragments were removed using the Picard 'MarkDuplicates' script with the options: Remove_Duplicates = true Validation_stringency = lenient (http://broadinstitute.github.io/picard/).

Paired-end reads were separated, centered on Tn5 cut sites, and trimmed to 10 bp using a custom in-house script (Source Code File 1). Peaks were called using the MACS2 'callpeak' script (*Zhang et al., 2008*) with options: -B –keep-dup all –nomodel –nolambda –shift −75 –extsize 150. Reads mapping to murine blacklisted regions and mitochondrial DNA were masked out of peak lists using the Bedtools 'intersect -v' script (*Quinlan and Hall, 2010*).

Replicates from each biological group were merged using the bedops 'merge' script to generate one high-confidence peak set for each of the four biological groups (GFP^high, GFP^low, *S1pr1* ECKO, *S1pr1* WT) (*Neph et al., 2012*). These four peak sets were then merged to generate a merged, consensus peak set of 123,473 peaks. For each replicate, reads covering consensus peak intervals were counted using the Bedtools 'coverage' script with the '-counts' option (*Quinlan and Hall, 2010*). The resultant count table was input to edgeR (*Robinson et al., 2010*) to determine differentially accessible peaks (DAPs).

DAPs were used as input for the HOMER 'findMotifsGenome.pl' script with the option '-size given' to identify motifs enriched in peaks with enhanced accessibility in either GFP$^{high}$, GFP$^{low}$, S1pr1 ECKO, or S1pr1 WT MAECs (Heinz et al., 2010).

Nucleotide-resolution coverage (bigWig) tracks were generated by first combining trimmed reads from each replicate, then inputting the resultant. bam files to the DeepTools (Ramírez et al., 2016) 'bamCoverage' script with options '—normalizeUsing RPGC –binSize 1'. Heatmaps of ATAC-seq reads within 600 bp of p65, NUR77, COUP-TFII, ATF1, GATA2, and GRE motifs were generated by centering coverage tracks on each motif identified in DAPs. These motifs were identified using the HOMER script 'annotatePeaks.pl' with the '-m -mbed' options. All heatmaps were generated using DeepTools and all genome browser images were captured using Integrative Genomics Viewer (Robinson et al., 2011).

## scRNA-seq analysis

1152 Fastq files (one per cell) were aligned to the GRCm38 (mm10) genome assembly using STAR with options –runThreadN 4 –outSAMstrandField intronMotif –twopassmode Basic. Bam files were input to the velocyto (http://velocyto.org/velocyto.py/) command-line script 'run-smartseq2' (La Manno et al., 2018). Expressed repetitive elments were downloaded from the UCSC genome browser and masked from analysis using the '-m' option of the 'run-smartseq2' script. The resultant table of read counts per transcript ('loom' file) was input to the PAGODA2 (https://github.com/hms-dbmi/pagoda2) R package for further analysis (Fan et al., 2016; La Manno et al., 2018). The details of our R code are provided in Source Code File 3.

After variance normalization, the top 3000 overdispersed genes were used for principal component analysis (PCA). An approximate k-nearest neighbor graph (k = 30) based on a cosine distance of the top 100 principal components was used for clustering. Clusters were determined using the multilevel community detection algorithm. PCA results were plotted using the 'tSNE' embedding option of the PAGODA2 'r$getEmbedding' function. Heatmaps of gene expression embedded on hierarchical clustering, differential expression analyses, and expression of individual transcripts on the tSNE embedding were generated using the graphical user interface at http://pklab.med.harvard.edu/nikolas/pagoda2/frontend/current/pagodaLocal/. We generated the binary (.bin) file according to the Pagoda2 walkthrough: https://github.com/hms-dbmi/pagoda2/blob/master/vignettes/pagoda2.walkthrough.oct2018.md. This binary file (Supplementary file 7) can be uploaded to the graphical user interface for exploration of our dataset. We generated a file of cluster labels (for LEC, vEC, VSMC, aEC1, aEC2, aEC3, aEC4, aEC5, aEC6 as well as all other cluster grouping used for analysis), which can also be uploaded to the graphical user interface for visualization of these clusters (Supplementary file 8).

## Immunohistochemistry

Mice were euthanized as described above, then perfused through the left ventricle with 5 mL PBS immediately followed by 10 mL ice-fold 4% paraformaldehyde (PFA) in PBS. The left ventricle was then perfused with 6 mL PBS. After aorta dissection, remaining fat tissue was removed with the aorta suspended in PBS in a polystyrene dish. For sectioning, intact thoracic aortae were additionally post-fixed in 4% PFA at 4° for 10 min, then briefly washed with PBS three times. Aortae were then cryo-protected in 30% sucrose in PBS for 2 hr at 4°C, embedded in a 1:1 mixture of 30% sucrose PBS: OCT over dry ice, then sectioned at 14 µM intervals using a cryostat (Leica Biosystems). E11.5 embryos were fixed in 4% PFA at 4° for 1 hr, briefly washed with PBS three times, cryoprotected in 30% sucrose in PBS for 24 hr at 4°C, and embedded as described above for sagittal sectioning with a cryostat.

For en face preparations, fine scissors were used to cut the aorta open and expose the endothelium for downstream flat-mount preparation. Aortae were placed into 24-well tissue culture plates and permeabilized in 0.5% Triton X-100/PBS (PBS-T) for 30 min on a room-temperature orbital shaker, then blocked in blocking solution (1% BSA (Fisher Scientific, BP1605), 0.5% normal donkey serum (Sigma-Aldrich, D9663)) in PBS-T for 1 hr. Primary antibody incubations were carried out overnight in blocking solution, followed by detection with secondary antibodies. Primary antibodies used were goat anti-VE-cadherin (1:300, R and D systems, AF1002), goat anti-Fibrinogen (Accurate Chemical, YNGMFBG), rat anti-Ki-67 (eBioscience, 14-5698-80), rabbit anti-LYVE1 (1:300, 103-PA50AG,

ReliaTech), goat anti-LYVE1 (1:300, R and D systems, AF2125), goat anti-VEGFR3 (1:200, R and D systems), rat anti-ITGA6 (1:200, BioLegend, 313602) goat anti-ALPL (1:200, R and D systems, AF2910), biotinylated mouse-anti TSP1 (1:200, ThermoFisher, MA5-13395), Alexa Fluor 488-conjugated mouse anti-CLDN5 (1:100, Invitrogen, 352588), rabbit anti-FSP1 (1:300, MilliporeSigma, 07–2274), goat anti-CD31 (1:300, R and D systems, AF3628), rat anti-CD31 (1:100, HistoBiotec, DIA-310), rabbit anti-LEF1 (1:200, Cell Signaling Technologies, 2230), rat anti-VCAM1 (1:200, Millipore-Sigma, CBL1300), goat anti-NOG (1:200, R and D systems, AF719), goat anti-CCL21 (1:200, R and D systems, AF457), goat anti-PROX1 (1:100, R and D systems, AF2727), mouse anti-ASMA-Cy3 (1:300, Sigma-Aldrich, C6198). H2B-GFP fluorescence was dected without an antibody for each experiment except imaging of mesenteric vessels, for which chicken anti-GFP (1:500, ThermoFisher, A10262) was used. Following primary antibody incubation, aortae were washed three times in PBS-T for 20 min each, then incubated with secondary antibodies in blocking solution at room temperature for 90 min. Donkey anti-rat, anti-rabbit, anti-chicken, and anti-goat secondary antibodies were purchased from ThermoFisher or Jackson ImmunoResearch as conjugated to Alexa Fluor 405, 488, 546, 568, 594, or 647. TSP1 was detected with streptavidin from Jackson Immunoresearch conjugated to Alexa Fluor 594 or 647. After secondary antibody incubation, aortae were washed in PBS-T for 20 min four times, then once in PBS, then mounted in mounting reagent (ProLong Gold, Invitrogen) on a slide with the tunica intima in contact with the coverslip. A textbook-sized weight was placed on top of the coverslip for 1 min before sealing with nail polish.

For staining of mesenteric vessels, mice were euthanized as described above then perfusion-fixed with 1% PFA, followed by post-fixation of mesenteries for 1 hr in 4% PFA. Staining was performed as described for aorta whole-mount preparations with the following exceptions: PBS-T contained 0.2% Triton X-100, blocking solution contained 3% BSA and 5% normal donkey serum in PBS-T, and secondary antibody incubation occurred overnight.

## Confocal microscopy and image analysis

Images were acquired using a Zeiss LSM810 confocal microscope equipped with an Plan-Apochromat 20x/0.8 or a Plan-Apochromat 40x/1.4 oil DIC objective. Images were captured using Zen2.1 (Zeiss) software and processed with Fiji (NIH). Zen2.1 software was used to threshold GFP signal and manually count GFP+ nuclei per field (*Figure 6*). Fiji was used to quantify GFP signal over PROX1+ASMA+ and PROX1+ASMA- areas (*Figure 8*), TSP1 signal normalized to total VEC pixels per field (*Figure 7F*), as well aso for quantification of branch point and non-branch point GFP+ nuclei (*Figure 7*) using the watershed segmentation tool. Figures were assembled in Adobe Illustrator.

## S1P analysis

Plasma S1P was extracted as previously decribed (*Frej et al., 2015*) with minor modification. Plasma aliquots (5 or 10 µL) were first diluted to 100 µL with TBS Buffer (50 mM Tris-HCl pH 7.5, 0.15 M NaCl). S1P was extracted by adding 100 µL precipitation solution (20 nM D7-S1P in methanol) followed by 30 s of vortexing. Precipitated samples were centrifuged at 18,000 rpm for 5 min and supernatant were transferred to vials for LC-MS/MS analysis (see below).

C18-S1P (Avanti Lipids) was dissolved in methanol to obtain a 1 mM stock solution. Standard samples were prepared by diluting the stock in 4% fatty acid free BSA (Sigma-Aldrich) in TBS to obtain 1 µM and stored at −80°C. Before analysis, the 1 µM S1P solution was diluted with 4% BSA in TBS to obtain the following concentrations: 0.5 µM, 0.25 µM, 0.125 µM, 0.0625 µM, 0.03125 µM, 0.0156 µM, and 0.0078 µM. S1P in diluted samples (100 µL) were extracted with 100 µL of methanol containing 20 nM of D7-S1P followed by 30 s of vortexing. Precipitated samples were centrifuged at 18,000 rpm for 5 min and the supernatants were transferred to vials for LC-MS/MS analysis. The internal deuterium-labeled standard (D7-S1P, Avanti Lipids) was dissolved in methanol to obtain a 200 nM stock solution and stored at −20°C. Before analysis, the stock solution was diluted to 20 nM for sample precipitation.

## LC-MS/MS S1P measurement and data analysis

The samples were analyzed with Q Exactive mass spectrometer coupled to a Vanquish UHPLC System (Thermo Fisher Scientific). Analytes were separated using a reverse phase column maintained at 60°C (XSelect CSH C18 XP column 2.5 µm, 2.1 mm X 50 mm, Waters). The gradient solvents were as

follows: Solvent A (water/methanol/formic acid 97/2/1 (v/v/v)) and Solvent B (methanol/acetone/water/formic acid 68/29/2/1 (v/v/v/v)). The analytical gradient was run at 0.4 mL/min from 50–100% Solvent B for 5.4 min, 100% for 5.5 min, followed by one minute of 50% Solvent B. A targeted MS2 strategy (also known as parallel reaction monitoring, PRM) was performed to isolate S1P (380.26 m/z) and D7-S1P (387.30 m/z) using a 1.6 m/z window, and the HCD-activated (stepped CE 25, 30, 50%) MS2 ions were scanned in the Orbitrap at 70 K. The area under the curve (AUC) of MS2 ions (S1P, 264.2686 m/z; D7-S1P, 271.3125 m/z) was calculated using Skyline (*MacLean et al., 2010*).

Quantitative linearity was determined by plotting the AUC of the standard samples (C18-S1P) normalized by the AUC of internal standard (D7-S1P); (y) versus the spiked concentration of S1P (x). Correlation coefficient ($R^2$) was calculated as the value of the joint variation between x and y. Linear regression equation was used to determined analyte concentrations.

## Acknowledgements

We thank Boston Children's Hospital and Harvard Stem Cell Institute Flow Cytometry Research Facility and Harvard Medical School Biopolymers Facility for technical assistance. We also thank members of the Hla laboratory for critical comments on the project and Dr. Sylvain Galvani for technical advice regarding mouse aorta dissection. This work is supported by NIH grants (R35-HL135821 to TH), Fondation Leducq Transatlantic Network grant (SphingoNet) to TH, CB, EC and RLP, and the intramural research program of NIDDK intramural program grants (RLP). MVL was the recipient of a postdoctoral fellowship from the Fonds de Recherche en Santé du Québec, AN is the recipient of a fellowship from the Marie Curie Prestige program, HN is the recipient of a fellowship from the Higher Education Commission, Pakistan, and AK was the recipient of a postdoctoral fellowship from the American Heart Association.

## Additional information

### Competing interests

Timothy Hla: received grant support from ONO Pharmaceuticals (2015-2018), has filed patent applications on ApoM, ApoM-Fc and HDL containing ApoM (US 62/545,629, PCT/US2018/000202, US 62/744,903, PCT/US2019/055831, US16/326,089, CA3034243, CN201780056922.6, JP 2019-530362, EPO 17851271.1), and has consulted for the following commercial entities: Astellas, Steptoe and Johnson, Gerson Lehrman Group Council, Janssen Research & Development, LLC, and Sun Pharma advanced research group (SPARC). The other authors declare that no competing interests exist.

### Funding

| Funder | Grant reference number | Author |
| --- | --- | --- |
| National Heart, Lung, and Blood Institute | R35 HL135821 | Timothy Hla |
| Fondation Leducq | SphingoNet Transatlantic Network Grant | Richard L Proia<br>Eric Camerer<br>Christer Betsholtz<br>Timothy Hla |
| National Institute of Diabetes and Digestive and Kidney Diseases | Intramural program support | Richard L Proia |
| Fonds de Recherche en Santé du Québec | Postdoctoral fellowship | Michel V Levesque |
| Marie Curie Prestige Program | Fellowship | Anja Nitzsche |
| Higher Education Commission, Pakistan | | Hira Niazi |
| American Heart Association | | Andrew Kuo |

The funders had no role in study design, data collection and interpretation, or the decision to submit the work for publication.

## Author contributions
Eric Engelbrecht, Data curation, Software, Formal analysis, Validation, Investigation, Visualization, Methodology; Michel V Levesque, Conceptualization, Formal analysis, Validation, Investigation, Methodology; Liqun He, Data curation, Software, Validation, Visualization; Michael Vanlandewijck, Investigation, Methodology; Anja Nitzsche, Andrew Kuo, Investigation; Hira Niazi, Investigation, Writing - review and editing; Sasha A Singh, Methodology, Project administration; Masanori Aikawa, Resources, Supervision, Funding acquisition, Project administration; Kristina Holton, Software, Formal analysis; Richard L Proia, Conceptualization, Resources, Funding acquisition, Project administration; Mari Kono, William T Pu, Resources, Methodology; Eric Camerer, Resources, Supervision, Funding acquisition, Investigation; Christer Betsholtz, Conceptualization, Resources, Supervision, Funding acquisition, Project administration; Timothy Hla, Conceptualization, Resources, Supervision, Funding acquisition, Methodology, Project administration

## Author ORCIDs
Anja Nitzsche (iD) http://orcid.org/0000-0003-0567-6790
Masanori Aikawa (iD) http://orcid.org/0000-0002-9275-2079
Richard L Proia (iD) http://orcid.org/0000-0003-0456-1270
Mari Kono (iD) http://orcid.org/0000-0003-2447-4350
William T Pu (iD) http://orcid.org/0000-0002-4551-8079
Eric Camerer (iD) http://orcid.org/0000-0002-6271-7125
Timothy Hla (iD) https://orcid.org/0000-0001-8355-4065

## Ethics
Animal experimentation: This study was performed in strict accordance with the recommendations in the Guide for the Care and Use of Laboratory Animals of the National Institutes of Health. All of the animals were handled according to approved institutional animal care and use committee (IACUC) protocols (#16-10-3297) of the Boston Children's Hospital. All surgery was performed under sodium pentobarbital anesthesia, and every effort was made to minimize suffering.

## Decision letter and Author response
Decision letter https://doi.org/10.7554/eLife.52690.sa1
Author response https://doi.org/10.7554/eLife.52690.sa2

# Additional files

## Supplementary files
• Source code 1. These files include scripts used to center ATAC-seq paired-end reads on Tn5 cut sites.

• Source code 2. STAR code used for mapping of scRNA-seq. fastq files.

• Source code 3. Pagoda2 R code used for scRNA-seq analysis.

• Supplementary file 1. edgeR results from bulk RNA-seq differential gene expression analysis for GFP$^{high}$ vs. GFP$^{low}$ and *S1pr1* ECKO vs. *S1pr1* WT MAEC comparisons. RSEM-generated FPKM values for all annotated transcripts are also included for each biological replicate.

• Supplementary file 2. Ingenuity Pathway Analysis results from bulk RNA-seq comparisons of GFP$^{high}$ vs. GFP$^{low}$ and *S1pr1* ECKO vs. *S1pr1* WT MAECs.

• Supplementary file 3. Bulk ATAC-seq of GFP$^{high}$ vs. GFP$^{low}$ and *S1pr1* ECKO vs. *S1pr1* WT MAECs. edgeR results from differential accessibility analysis are included, as well as the genomics coordinates of differentially accessible peaks and the motif analyses (HOMER) results shown in *Figure 2C–F*.

- Supplementary file 4. scRNA-seq of GFP$^{high}$ and GFP$^{low}$ MAECs. Tabs include transcripts enriched and depleted in each cluster with details at the top of each page. Also included are 2 tabs of transcripts that are commonly enriched or depleted in LEC/vEC/aEC1 (referred to in *Figure 3—figure supplement 4*), as well as a tab detailing the intersection of aEC1-enriched transcripts and those differentially expressed in the *S1pr1* ECKO dataset (*Figure 6A and B*).

- Supplementary file 5. Tabs include Gene Set Enrichment Analysis (GSEA) results using transcripts enriched in each cluster as inputs. Also included is the complete list of sphingolipid-related genes queried (referred to in *Figure 3—figure supplement 4A*). The list of transcription factors is limited to those identified as aEC1-enriched and –depleted that also had minimal count thresholds after Pagoda2 filtering during hierarchical differential expression analysis.

- Supplementary file 6. Detailed intersections of *S1pr1* ECKO up- and down-regulated transcripts according to their cluster assignments from scRNA-seq.

- Supplementary file 7. Binary '.bin' file that can be uploaded to the graphical user interface: http://pklab.med.harvard.edu/nikolas/pagoda2/frontend/current/pagodaLocal/ for examination of our GFP$^{high}$ and GFP$^{low}$ MAEC scRNA-seq data.

- Supplementary file 8. The labels for each scRNA-seq cluster, which can also be uploaded at: http://pklab.med.harvard.edu/nikolas/pagoda2/frontend/current/pagodaLocal/.

- Transparent reporting form

## Data availability
Sequencing data and processed files have been deposited in GEO under the accession GSE139065.

The following dataset was generated:

| Author(s) | Year | Dataset title | Dataset URL | Database and Identifier |
|---|---|---|---|---|
| Hla T | 2019 | Sphingosine 1-phosphate-regulated transcriptomes in heterogenous arterial and lymphatic endothelium of the aorta | https://www.ncbi.nlm.nih.gov/geo/query/acc.cgi?acc=GSE139065 | NCBI Gene Expression Omnibus, GSE139065 |

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
