## [Decision Letter]

**Acceptance summary:**

We are particularly excited by the combination of an elegant reporter system for SIPR1 signaling in vivo with RNA profiling and use of genetic manipulations to further our understanding of endothelial cell heterogeneity in general, and the role of SIPR signaling in this heterogeneity. It is very useful that your work provides in vivo spatial resolution to correlate with the profiling, and that novel links between inflammatory signatures and SIPR status have been revealed. This work is technically rigorous and has strong biological impact.

**Decision letter after peer review:**

Thank you for submitting your article "Sphingosine 1-phosphate-regulated transcriptomes in heterogenous arterial and lymphatic endothelium of the aorta" for consideration by *eLife*. Your article has been reviewed by three peer reviewers, and the evaluation has been overseen by a Reviewing Editor and Didier Stainier as the Senior Editor. The following individual involved in review of your submission has agreed to reveal their identity: Luisa Iruela-Arispe (Reviewer #3).

The reviewers and Reviewing Editor have discussed the reviews with one another, and we are impressed with the quantity and quality of the work and the findings that SIP-signaling appears to be heterogeneous and composed of both ligand-dependent and ligand-independent processes. We do have some concerns that we believe can be addressed with more data analysis and/or text edits, and we have drafted this decision to help you prepare a revised submission. There are several questions around the reporter line used, and some questions around the bioinformatics, along with several minor points. We look forward to seeing a revised version of this impactful work.

Summary:

This study uses a combination of transcriptome sequencing, chromatin accessibility, and immunostaining to investigate the role of sphingosine 1-phosphate receptor (S1PR1) signaling in aortic endothelial cells. They combine a mouse reporter line for S1PR1-β-arrestin coupling, and knockout lines for S1PR1 and its ligand, S1P, to identify the transcriptional effects of S1PR1 signaling and modulation of its activity by S1P. The data demonstrate that S1PR signals in vivo through canonical (ligand-dependent) and non-canonical (ligand-independent) manners in distinct subsets of endothelial cells. The non-canonical pathway appears restricted to a very interesting subpopulation of cells confined to branchpoints of the aorta. The authors previously showed ligand-independent signaling (via the S1PR1/β-arrestin mouse) in the lesser curvature and intercostal branch points, and here the authors characterize in detail the impact of both modes of signaling on the transcriptome of endothelial cells using state of the art techniques and animal models. The chromatin status of different cell populations was interrogated using ATACseq. The findings generally are well supported by the results provided, and the manuscript is carefully crafted with impressive data volume and overall quality.

Essential revisions:

1) The major revision point centers around the reporter mouse line used. All reviewers agreed that it was important to better characterize the reporter, and how it reflects (or does not) SIP signaling. It seems that this issue is likely to be addressable by re-analysis of some of the data and/or text edits.

Some of the questions raised around this point are:

– Is there published work that quantifies how concordant actual β-arrestin recruitment is with GFP expression?

– How likely is it that the reporter is an underestimate or an overestimate of activity due to persistence of *H2B-GFP*? Suggest to better describe data on reporter dynamics, since *H2B-GFP* is usually very stable and may be present long after signaling activation. S1P deletion removed inter branch reporter signal; it would be helpful to understand how long it takes to degrade the reporter.

– Suggest describe in more detail for each cell population what the data suggests happens to signaling in the reporter line, i.e. is it reporting receptor activation or chronic down-regulation? In Figure 1C, the greatest number of shared genes is between the S1PR1 ko and the GFP^high^. Does this suggest that GFP^high^ cells are reporting no signaling? Maybe this is why S1P knockout does not abolish branchpoint GFP. It could be that GFP reports chronic endocytosis and receptor down-regulation by shear stress at branch points, so that it does not respond to S1P.

– Please explain and discuss to what degree the S1PR1-GS signaling reporter allele, which is based on a knock-in targeting strategy, functions as a null or a hypomorphic allele. It is unclear whether the line is a good tool to isolate cells for transcriptomic analysis, as loss of S1PR1 function might affect transcriptional regulation in these cells.

– Without losing focus, can the authors elaborate a bit more on the flow implications of the ligand-independent pathway? The authors previously published on the role of flow in canonical *S1pr1* (Jung et al., 2012), and here some discussion of the impact of turbulent/disturbed flow on the ligand-independent induction of S1pr1 is warranted.

2) The reviewers also asked for some clarification of the presented bioinformatics data:

– Please provide some validation of FACS strategy in Figure 1B. The GFP-low and GFP^high^ gates overlap quite a bit. Can GFP transgene expression be included in the sequencing analysis, to confirm that the GFP^low^ gated cells have no expression? Alternatively, qPCR or immunostaining of cells from the sort.

– Figure 1E. Please specify which genes are downregulated in GFP^high^ and upregulated in S1PR1-ECKO, and what is their significance? Especially important since there is large overlap between GFP^high^ and S1PR1-ECKO enriched transcripts.

– GFP signal outside of branch points – please clarify distribution of GFP expression outside of branch points in the aorta. Figure 1A, Figure 5B, Figure 5—figure supplement 2A, Figure 6E, Figure 7B, look different in terms of the amount of "distal" GFP+ cells not directly at the branch points; Figure 7D – more time points needed to connect aging and reporter expression at non-branch points.

---

## [Author Response]

Essential revisions:1) The major revision point centers around the reporter mouse line used. All reviewers agreed that it was important to better characterize the reporter, and how it reflects (or does not) SIP signaling. It seems that this issue is likely to be addressable by re-analysis of some of the data and/or text edits.Some of the questions raised around this point are:– Is there published work that quantifies how concordant actual β-arrestin recruitment is with GFP expression?

Thank you for raising this important point regarding S1PR1-GFP signaling (S1PR1-GS) reporter mice.

Kono M. used the original Tango strategy (Barnea et al., 2008) to design the *S1pr1* knock-in allele. Barnea et al. showed that transfection of GPCRtTA and β-arrestin-TEV fusions into cells is a reliable method of monitoring activation of a specific GPCR. Activation of most GPCRs is followed by a β-arrestin-mediated desensitization step, which is recorded in the Tango system by expression of a tTA-responsive reporter gene. Banea G. et al. described this system as providing “a specific and quantitative measure of ligand-induced interaction between GPCR and arrestin”.

More specifically for the *S1pr1* Tango system, Kono et al., 2014, used MEF cells derived from S1PR1GS mice, which harbor germline *S1pr1*^ki/+^ and *H2B-GFP* alleles, and obtained an S1P EC_50_ similar to the literature, which is ~ 43 nM. Our recent publication confirmed the results of (Swendeman S.L. et al. 2017 Sci Signal. 2017 Aug 15;10(492)).

To address *H2B-GFP* expression that is independent of S1PR1 (i.e. negative control), we stained and imaged the *H2B-GFP* reporter mouse that lacks *S1pr1*-tTA-IRES-β-arrestin-TEV knock-in alleles (*S1pr1*^+/+^*H2B-GFP*). This analysis indicated that the basal level of the *H2B-GFP* signal varies from undetectable to background level, which is orders of magnitude lower compared to the signal in aortae of S1PR1-GS (*S1pr1*^ki/+^*H2B-GFP*) mice (Figure 1A). Similar data were observed in mesenteric lymphatic vessels (Figure 8C). Therefore, based on our data and the previous studies, we believe that the GFP expression is representative of the β-arrestin recruitment to S1PR1.

Zhou et al., 2017, used the Tango system in vitro for studies of rhodopsin, the B2-adrenergic and vasopressin-2 receptors. They show that specific serine/threonine à alanine mutations in the C-terminal tail resulted in loss of reporter expression. The mutated residues were predicted to mediate interactions with β-arrestin, and their experiments have helped to elucidate the “phosphorylation codes” on GPCRs that mediate arrestin recruitment.

These points have been noted in the revision.

– How likely is it that the reporter is an underestimate or an overestimate of activity due to persistence of H2B-GFP? Suggest to better describe data on reporter dynamics, since H2B-GFP is usually very stable and may be present long after signaling activation. S1P deletion removed inter branch reporter signal; it would be helpful to understand how long it takes to degrade the reporter.

The *H2B-GFP* signal likely underestimates the recruitment of ß-Arrestin to S1PR1 for the following reasons. In any cells of the mouse we use, there are always 1 allele of *S1pr1* and 2 alleles of ß-arrestin that are wild type (no fusion with the tango system elements). If we assume that both *S1pr1* alleles (*wt* and *S1pr1*-tTA) generate proteins with similar properties, 50% of the ß-arrestin recruitment to S1PR1 would involve a wild type receptor and generate no GFP expression, thus diluting the actual signal output. There is also a delay of about 6 hours between activation of the receptor and the observation of GFP in the nucleus (Kono et al., 2014). Finally, in cells were the receptor undergoes recycling to the cell membrane after stimulation, phosphorylation and β-arrestin dependent internalization, the tango will only record one event of β-arrestin recruitment. Subsequent activation of S1PR1 after cleavage of the S1PR1-tTA fusion, would not lead to increase signal of the *H2B-GFP* reporter.

In addition, the *H2B-GFP* protein half-life of (~ 6 hours) (Corish et al. Protein Eng. 1999 Dec;12(12):1035-40.) and slow turn-over of cells that are *H2B-GFP*+ in vivo(~ 24 days) have been reported (Waghmare et al., 2008). Thus, to make sure that we accurately estimate the plasma S1P-dependent activation of the reporter, we waited over 6 months between the beginning of the tamoxifen treatments and the harvest dates. Therefore, it is likely that the remaining GFP signal at branch point EC is from ligand-independent receptor activation.

– Suggest describe in more detail for each cell population what the data suggests happens to signaling in the reporter line, i.e. is it reporting receptor activation or chronic down-regulation? In Figure 1C, the greatest number of shared genes is between the S1PR1 ko and the GFP^high^. Does this suggest that GFP^high^ cells are reporting no signaling? Maybe this is why S1P knockout does not abolish branchpoint GFP. It could be that GFP reports chronic endocytosis and receptor down-regulation by shear stress at branch points, so that it does not respond to S1P.

The *S1pr1* signaling mouse reports ß-Arrestin recruitment to S1PR1. This event leads to acute signaling via the Gi protein as well as receptor endocytosis and downregulation. Thus, the GFP reporter reflects both positive and negative aspects of S1PR1 signaling.

Since the branch point-specific (aEC1) cells exhibit ligand-independent reporter activity, cell-intrinsic mechanisms are likely responsible. In contrast, aEC2 and LEC populations exhibit ligand-dependent reporter activity.

Bulk transcriptomics indicated that GFP^high^ ECs show more similarity with *S1pr1-ECKO* that with *WT* ECs. We note that 97 genes from the 258 *S1pr1-ECKO* upregulated genes were enriched in at least one of the clusters identified by scRNA-seq (Figure 9). Moreover, 80% of them (78 genes) were enriched in the LEC cluster. The cells forming the LEC cluster were 96% GFP^high^ cells. Taken together, *H2B-GFP* expression in LECs is may be downstream of a “desensitization-like mechanism” rather than positive S1PR1 signaling.

In addition, shear forces (laminar vs. disturbed flow) could participate in the reporter expression. However, the expression of our aEC1 cluster markers such has *Itga6* are likely expressed independently of biomechanical signaling since its expression pattern does not fully overlap with classic disturbed shear responsive genes such as *Vcam1*.

– Please explain and discuss to what degree the S1PR1-GS signaling reporter allele, which is based on a knock-in targeting strategy, functions as a null or a hypomorphic allele. It is unclear whether the line is a good tool to isolate cells for transcriptomic analysis, as loss of S1PR1 function might affect transcriptional regulation in these cells.

We included three new sets of data (Figure 1—figure supplement 1A, B and C) which suggest that *S1pr1*^ki/+^ mice do not exhibit evidence of hypomorphism. However, *S1pr1*^ki/ki^ homozygous mice exhibit evidence of partially penetrant hypomorphism. Firstly, *S1pr1*^ki/+^ animal are born at the expected Mendelian frequency. Secondly, the analysis of the levels of circulating lymphocytes are unchanged between *S1pr1*^+/+^ and *S1pr1*^ki/+^ animals. Thirdly, there is no increase in lung leakage between *S1pr1*^+/+^ and *S1pr1*^ki/+^ animals. Because all our experiments were conducted on animals carrying a single copy of the “*S1pr1-tDA-IRESβ-Arrestin-TEV*” allele and a wildtype allele, we believe that our data represent biologically relevant transcriptomic information. Additionally, since we sorted the GFP^high^ and GFP^low^ MAECs from the same animals (both subpopulations have the same genotype *S1pr1*^ki/+^) the difference observed between them is dependent on receptor signaling. Our data are further supported by the fact that we were able to demonstrate the existence of unique MAEC clusters (aEC1 for example) in wildtype mice (Figure 5—figure supplement 1A).

– Without losing focus, can the authors elaborate a bit more on the flow implications of the ligand-independent pathway? The authors previously published on the role of flow in canonical S1pr1 (Jung et al., 2012), and here some discussion of the impact of turbulent/disturbed flow on the ligand-independent induction of S1pr1 is warranted.

Our data suggest that the unique transcriptome profile of aEC1 is likely the result of cell lineage and not flow signaling. First, the transcriptome of the branch point ECs (aEC1) do not present the classical proinflammatory profile expected from ECs exposed to disturbed flow. Second, region of the aorta with well characterized exposure to disturbed flow (lesser curvature) do not stain positive for ITGA6 protein (aEC1 marker). However, shear forces may contribute to a specific phenotypic alteration in these cells.

2) The reviewers also asked for some clarification of the presented bioinformatics data:– Please provide some validation of FACS strategy in Figure 1B. The GFP^low^ and GFP^high^ gates overlap quite a bit. Can GFP transgene expression be included in the sequencing analysis, to confirm that the GFP^low^ gated cells have no expression? Alternatively, qPCR or immunostaining of cells from the sort.

In our FACS strategy, we specifically used gates that would exclude MAECs with intermediate GFP signal (see Figure 1B) so that GFP^low^ and -high cells will *not* overlap. The FACS analysis of MAECs from mice lacking the *H2B-GFP* allele (no GFP expression) (Figure 1—figure supplement 2A) shows that all cells are included in the GFP^low^ gate and there are no cells in the GFP^high^ gate area. To clarify the gating strategy, we edited the Figure 1B, Figure 1—figure supplement 2A and their respective legend. As suggested by the reviewers, we also have included GFP transgene expression data. The GFP^high^ MAECs showed an ~20-fold increase in *eGFP* transcripts relative to GFP^low^ MAECs in the bulk RNAseq data (Figure 1—figure supplement 2B). Similarly, the scRNAseq data shows detectable level of *eGFP* almost exclusively in cells from the GFP^high^ population (green bars) (Figure 3—figure supplement 1A).

– Figure 1E- Please specify which genes are downregulated in GFP^high^ and upregulated in S1PR1-ECKO, and what is their significance? Especially important since there is large overlap between GFP^high^ and S1PR1-ECKO enriched transcripts.

We present all overlapping genes from Figure 1C and 1E in the Supplementary file 1 in the tab named “GFP-ECKO_Intersec”. It was not clearly referenced in the previous version of the manuscript. We edited the manuscript to help the reader to find the information more easily. For the convenience of the reviewer, we provide here the list of the GFP^low^ enriched genes also upregulated in the *S1pr1-ECKO: Amac1, F2rl3, Fmod, Nod2, Plxnc1, Prnd, Sfrp2, Spn*. These 8 genes were not found has enriched or depleted in any of the 9 clusters that we have identified.

– GFP signal outside of branch points – please clarify distribution of GFP expression outside of branch points in the aorta. Figure 1A, Figure 5B, Figure 5—figure supplement 2A, Figure 6E, Figure 7B, look different in terms of the amount of "distal" GFP+ cells not directly at the branch points; Figure 7D – more time points needed to connect aging and reporter expression at non-branch points.

We thank the reviewer for this question and appreciate the opportunity to provide clarification (see below). Indeed, non-branch point GFP+ EC exhibit varying levels of GFP expression, from near-background but distinct nuclear expression up to the high levels we observe around the circumference branch points. GFP^low^ EC may appear as GFP- when the image is merged with VE-Cadherin and/or other stains. In consideration of space, we chose to show the GFP channel merged with other channels for several panels that the reviewer mentioned (Figure 1A, Figure 5B, Figure 7B).

Figure 1A – relatively zoomed out, no isolated GFP channel. This makes the heterogeneous low GFP signal hard to see.

Figure 5B – focused only on the branch point, no isolated GFP channel. See above.

Figure 5—figure supplement 2A – isolated GFP channel is shown.

Figure 6E –these mice were 7 months old and therefore are different from the majority of mice used for imaging (3-4 months old). Also, their background is different from normal S1PR1-GS mice (these are *Sphk2*-null mice).

Figure 7B – the GFP channel is hard to see in the merge, so same case as Figure 1A.

To thoroughly address branch point and non-branch point GFP+ EC frequency, we imaged large fields that span the diameter of the aorta and include at least 4 branch points (Figure 7—figure supplement 2). We show the GFP channel merged with VE-Cadherin (VEC) and in isolation. The pattern of these non-branch GFP+ ECs in 3-month-old mice appears heterogeneous. In young adult mice, which were used for the majority of this study, we often see relatively low GFP+ EC frequency at regions most distal from the branch (bottom 10 rows and top 5 rows of cells in Figure 7—figure supplement 2C).

The revised manuscript includes quantification of GFP+ EC at branch point and non-branch point regions (Figure 7D) in wide fields (as shown in Figure 7—figure supplement 2) from P6, P12, young adult, and 15-month-old mice. These large fields should facilitate readers’ appreciation for the distribution of *H2B-GFP* expression throughout the aorta at P6, P12, 3 months and 15 months (Figure 7—figure supplement 2B, C and D).

Similar to the expansion of non-branch point (aEC2-like) EC, we show that TSP1 (an aEC2 marker) expression increases dramatically with age (between P6 and young adult stages), supporting the concept that the aEC2 cells include a population of activated MAECs that expands to expand with age. We have edited the manuscript to take these elements into account.

We extended our analysis to embryonic stages by immunohistochemical analysis of ITGA6, CD31, TSP1, and LYVE1 in E11.5 S1PR1-GS embryos. In contrast to the adult mouse aorta, embryonic dorsal aortae showed ubiquitous ITGA6 expression but were devoid of TSP1 expression. In Figure 7 and Figure 7—figure supplement 3E we provide evidence demonstrating that other aEC1-enriched transcripts, such as *Igf2, Slc6a6, Flt1*, and *Lef1* are expressed in the dorsal aorta. In contrast, transcripts depleted from aEC1 (*Sod3, Vcam1, Pcolce2, Frzb*) are expressed at low levels or absent in embryonic EC.